# Microbiome Engineering for Sustainable Rice Production: Strategies for Biofertilization, Stress Tolerance, and Climate Resilience

**DOI:** 10.3390/microorganisms13020233

**Published:** 2025-01-22

**Authors:** Israt Jahan Misu, Md. Omar Kayess, Md. Nurealam Siddiqui, Dipali Rani Gupta, M. Nazrul Islam, Tofazzal Islam

**Affiliations:** 1Institute of Biotechnology and Genetic Engineering (IBGE), Bangabandhu Sheikh Mujibur Rahman Agricultural University, Gazipur 1706, Bangladesh; isratjahan.mishu71p@gmail.com (I.J.M.); kbdkayess@gmail.com (M.O.K.); drgupta80@gmail.com (D.R.G.); 2Department of Biochemistry and Molecular Biology, Bangabandhu Sheikh Mujibur Rahman Agricultural University, Gazipur 1706, Bangladesh; nuralambmb@bsmrau.edu.bd; 3Centre for Plant and Soil Health, Regenerative Agri-Science Canada Inc., Winnipeg, MB R3T 5L2, Canada

**Keywords:** microbiome engineering, rice production, biofertilization, stress tolerance, microbiome-shaping genes, metagenomics, impact of global climate change, sustainable agriculture

## Abstract

The plant microbiome, found in the rhizosphere, phyllosphere, and endosphere, is essential for nutrient acquisition, stress tolerance, and the overall health of plants. This review aims to update our knowledge of and critically discuss the diversity and functional roles of the rice microbiome, as well as microbiome engineering strategies to enhance biofertilization and stress resilience. Rice hosts various microorganisms that affect nutrient cycling, growth promotion, and resistance to stresses. Microorganisms carry out these functions through nitrogen fixation, phytohormone and metabolite production, enhanced nutrient solubilization and uptake, and regulation of host gene expression. Recent research on molecular biology has elucidated the complex interactions within rice microbiomes and the signalling mechanisms that establish beneficial microbial communities, which are crucial for sustainable rice production and environmental health. Crucial factors for the successful commercialization of microbial agents in rice production include soil properties, practical environmental field conditions, and plant genotype. Advances in microbiome engineering, from traditional inoculants to synthetic biology, optimize nutrient availability and enhance resilience to abiotic stresses like drought. Climate change intensifies these challenges, but microbiome innovations and microbiome-shaping genes (M genes) offer promising solutions for crop resilience. This review also discusses the environmental and agronomic implications of microbiome engineering, emphasizing the need for further exploration of M genes for breeding disease resistance traits. Ultimately, we provide an update to the current findings on microbiome engineering in rice, highlighting pathways to enhance crop productivity sustainably while minimizing environmental impacts.

## 1. Introduction

Microbiome engineering (ME) presents a transformative approach to boosting agricultural productivity and enhancing climate resilience [1,2]. By harnessing the power of plant and soil microbial communities, ME optimizes ecosystem functions to promote plant growth and development [3,4]. This cutting-edge biotechnological strategy holds immense promise in addressing global food security challenges amid rapidly changing environmental conditions [5]. Symbiotic relationships with beneficial microbes’ boost plants’ resilience to biotic and abiotic stresses [6]. These interactions enhance plant health and productivity, which are essential for achieving sustainable agriculture [7,8]. The plant-associated microbiota encompasses the microbial communities inhabiting various niches, including epiphytes on plant surfaces, endophytes residing within plant tissues (both intercellular and intracellular), and microorganisms in the rhizosphere, endosphere, and phyllosphere [9]. Several promising bacterial genera, including *Rhizobium*, *Bacillus*, *Paraburkholderia*, *Delftia*, *Pseudomonas*, *Enterobacter*, *Lysobacter*, *Serratia*, *Alcaligenes*, *Burkholderia*, *Azotobacter*, *Azospirillum*, *Brevibacterium*, *Clostridium*, and others, have shown significant potential in enhancing plant growth, improving yields, and strengthening defence mechanisms for sustainable agriculture. [8,10,11,12]. Additionally, fungal communities in the rice mycorrhizosphere are typically dominated by genera like *Penicillium*, *Aspergillus*, *Talaromyces*, *Trichoderma*, and various arbuscular mycorrhizal fungi [13]. These microbiomes can be customized to specific crop–soil–environment combinations, providing a targeted approach to plant support. This symbiotic relationship fosters improved crop yields, soil health, and ecological balance.

Agriculture faces significant environmental issues in the modern era, including water scarcity, climate change, and declining soil health and water quality [14,15,16]. The United Nations (2019) projects that the global population will rise from 8.2 billion to 9.7 billion by 2050, creating the need for a 70–110% increase in food production to meet nutritional demands. Rice, being a staple food for over 3.5 billion people worldwide, will play a crucial role in addressing this need, particularly in developing countries [17,18]. The demand for rice is expected to rise continuously, being propelled by the steady growth of the global population [19,20]. The current use of agrochemicals and inorganic fertilizers has contributed to the degradation of soil health and water quality, dysbiosis of the microbiome, a decline in beneficial insect populations, a loss of biodiversity, the disruption of ecosystem functions, an increased vulnerability of rice plants to diseases, and an imbalance in microbial structures, causing irreversible harm to agriculture systems [21,22,23]. Only a limited percentage of applied chemical pesticides (about 1%) are reported to reach their intended targets due to volatilization, leaching, photolysis, and runoff [24]. With the combination of contemporary technology and environment-friendly management techniques, the application of the plant microbiome has opened up new opportunities for a sustainable, eco-friendly agricultural system [5,9,25,26,27].

Recently emerging plant microbiome research tools and techniques such as next-generation sequencing are essential for studying the whole-plant microbiota and its role in improving growth and productivity under stress [28]. The application of plant-beneficial bacteria improves the growth and yield of rice [29] by improving nutrient uptake and mitigating the impacts of both biotic and abiotic stresses [30,31,32]. Matsumoto et al. [33] identified that *Sphingomonas melonis* naturally confers resistance to seed-borne pathogens like *Burkholderia plantarii* in rice. For “biological resistance”, biopolymers like exopolysaccharides form a protective matrix around roots, while osmolytes, such as soluble sugars, amino acids, and organic solutes like proline and glycine betaine help plants withstand stress, as observed in drought-resistant plants and halophytes colonized by *Pseudomonas putida*, *Pantoea brenneri*, and *Acinetobacter calcoaceticus* [34]. These molecules help to maintain cell turgor during water deficit conditions and stabilize proteins and membranes, with soluble sugars preventing protein denaturation. Additionally, they enhance the plant’s antioxidant defence system by producing antioxidant enzymes and secondary metabolites such as 1-aminocyclopropane-1-carboxylate deaminase, β-aminobutyric acid, salicylic acid, and siderophores. The accumulation of osmoprotectants (ascorbate and proline) and phytohormones (ethylene and auxin) and the modulation of stomatal conductance further contribute to stress tolerance. Rice plants inoculated with arbuscular mycorrhizal fungi (AMF) exhibited an improved photosynthetic efficiency under saline and drought stress conditions via the direct uptake and translocation of water from the soil to the host plant via fungal hyphae, altering the water retention properties of soil and allowing for an improved adjustment of the osmotic potential of inoculated/treated plants [35,36]. This mutualistic association not only enhances water and nutrient availability but also strengthens the plant’s overall physiological resilience to environmental stresses. Additionally, microbes like *Pseudomonas* sp. and *P. brenneri* reduce oxidative stress through antioxidant enzymes and metabolites, and phytohormones or volatiles trigger induced systemic tolerance (IST) in plants, with 1-Aminocyclopropane-1-carboxylate (ACC) deaminase lowering ethylene levels to support stress resilience [37]. Microorganisms like *Pseudomonas*, *Bacillus*, *Trichoderma*, and AM fungi can trigger induced systemic resistance (ISR) in plants, which helps to protect them against various pathogens [38]. ISR activates defence-related genes, promoting the production of enzymes like phenylalanine ammonia-lyase, polyphenol oxidase, peroxidase, β-1,3-glucanase, and chitinase, along with the accumulation of reactive oxygen species [39]. Li et al. [40] reported that *Bacillus subtilis* KLBMPGC81 exhibited a superior biocontrol efficacy against *Magnaporthe oryzae* Guy11. In addition, entomopathogenic fungi use virulence factors like chitinase enzymes and mycotoxins to effectively infect insects [41].

PGPR (plant growth-promoting rhizobacteria) enhance nutrient absorption, particularly that of nitrogen and phosphorus which support rice growth [42]. Inoculating rice with Rhizobium bacteria resulted in approximately a 20% boost in dry weight, along with improvements in various plant growth parameters [43], and cyanobacteria can boost the rice grain yield by 11% and contributes 25–50% of the total nitrogen per hectare of fields each season [44]. *Azotobacter* and *Azospirillum* spp. have been shown to significantly enhance nitrogen availability in the soil [45]. The bacterial genera *Desulfobacteria*, *Entotheonella*, and *Algoriphagus*, along with members of *Anaerolineae* and *α-Proteobacteria* are essential for sulphur oxidation and reduction, nitrogen fixation, and phosphorus metabolism, enhancing nutrient availability and uptake in rice [46]. The inoculation of rice with various *Rhizobium* strains at different nitrogen levels resulted in a 4% to 19% increase in straw yield and an 8% to 22% increase in grain yield [47]. The Azolla–Anabaena system contributed 1.1 kg N ha^−1^ day^−1^, providing 20–40 kg N ha^−1^ to rice in 20–25 days [48]. Rice has demonstrated a favourable response to the application of *Pseudomonas* spp. as a PGPR, resulting in increased dry matter production and enhanced nitrogen uptake. Islam et al. [49] demonstrated that two rice probiotic bacteria, *Paraburkholderia fungorum* strain BRRh-4 and *Delftia* sp. strain BTL-M2, promoted the growth and yield of rice plants by 50%, reducing the recommended doses of N, P, and K fertilizers as their application produced an almost equivalent yield to treatment with 100% of the recommended doses of chemical fertilizers without the application of bacteria by increasing the availability and efficiency of the N, P, and K and by modulating the rice root-associated microbiome. Metagenomics analyses revealed that the application of these strains strikingly increased the diversity, structure, and composition of the associated beneficial bacteriome in both rice roots and rhizosphere soils and serve as eco-friendly alternatives (biofertilizers) to costly and hazardous synthetic fertilizers. These eco-friendly and cost-effective solutions not only improve plant yield, and defence systems but also protect against environmental stressors and offer advantages over harmful chemical fertilizers.

Microorganisms in agroecosystems facilitate communication among plants, with plant genes playing distinct roles in shaping (Ms genes) or responding (Mr genes) to microbial communities. Understanding these gene–microbe interactions could provide valuable insights into how specific gene sets influence and shape gene–phenotype relationships. Ms genes exert a significant influence on plant-associated microbial communities by regulating the production of specific small peptides and secondary metabolites (e.g., 4-hydroxycinnamic acid), and even through trans-kingdom miRNA transfer. This could allow plants to selectively recruit beneficial microbial communities to enhance their adaptability to various environments to support the host plant. Genome editing toolkits, such as CRISPR-Cas system, have rapidly advanced and are now indispensable genetic engineering tools for improving plant pathogen stress tolerance [50]. These approaches can reduce dependency on chemical inputs like fertilizer, pesticides and PGPRs, etc., contributing to more sustainable agriculture in the face of climate change. Challenges persist, including the complex nature of plant–microbe interactions, difficulties in ensuring long-term colonization and the stability of introduced microbes under field conditions, extensive field validation requirements, and regulatory hurdles. Public perceptions of genetic modifications also complicate adoption, highlighting the need for interdisciplinary research to understand plant–microbe dynamics and ensure effective application in agriculture. Some studies have established a clear connection between a particular plant chemical and the health of the leaf microbiome and suggests potential avenues for developing disease-resistant rice varieties through breeding.

Several studies and reviews have examined the rice microbiome [51,52,53]; however, few have delved into its diverse functions and potential applications in sustainable agriculture under real-world conditions. This review aims to critically analyze various aspects of the rice microbiome and outlines the following objectives: (1) to explore the diversity and dynamic structures of rice-associated microbiomes in the rhizosphere, phyllosphere, and endosphere, focusing on their roles in growth, nutrient acquisition, and stress resilience; (2) to examine signalling mechanisms in the rice rhizosphere and the contributions of microbes to growth, yield, and stress tolerance; (3) to highlight the role of metagenomics in mapping rice-associated microbial communities and their functional potential; (4) to discuss microbiome engineering strategies, including microbial inoculation, gene editing, and microbiome-shaping (M) genes to develop stress-resistant rice cultivars and enhancing climate resilience; and (5) to identify current challenges in microbial interactions, scaling microbiome applications, and integrating these strategies into sustainable rice production, while offering insights into future research directions.

## 2. Diversity and Functional Dynamics of Rice-Associated Microbiomes

Rice microbiomes exhibit diverse and dynamic structures, comprising intricate microbial communities that interact with host plants [49,54,55,56,57]. The biogeography of soil microbes significantly influences the composition of rice rhizosphere microbial communities across different planting locations [58]. Local environmental conditions, such as soil type and climate, directly determine the available microbial species pool and indirectly impact rhizosphere microbial communities by affecting plant and microbial physiology. Consequently, the relative contributions of plant genotype and planting location to shaping the rhizosphere microbiome remain a subject of ongoing research [59]. The rhizosphere microbial communities of Japonica rice and two hybrid rice breeds exhibit discernible structural differences, which are strongly correlated with rice yields [60]. The authors of this study found that planting location and soil type primarily drive the composition of rhizosphere microbial communities, while rice genotype has a secondary, but significant, role in selectively recruiting specific microbial taxa that influence microbial functions and yield potential. The rice rhizosphere, a specific microbial environment, is marked by varying levels of oxidation due to the roots’ release of oxygen and intensive water control. This unevenness may contribute to the limited presence of certain microbial groups [61].

Bacteria in the rice field significantly enhance the geochemical cycling of essential nutrient elements and also enhance the release of methane and other greenhouse gases that impact global warming as certain microbes are more tolerant to specific environmental stresses [62,63]. Rice fields are rich ecosystems which host diverse microorganisms [64,65], including nitrogen fixers, nitrifiers, and methanogens. These microbes play crucial roles in nutrient cycling, enhancing plant growth, and maintaining ecological balance [66]. The intricate interactions among these diverse components contribute to the overall health, nutrient cycling, and ecological balance within rice ecosystems [67]. The composition and interactions of rice microbiomes are influenced by soil type, rice genotype, agricultural practices, as well as biological, chemical, physical, and environmental factors [61,68,69].

Rice roots host three distinct microbial niches: the rhizosphere, the soil region around the roots influenced by root exudates and microbial activity; the rhizoplane, the root surface that serves as a habitat for various microorganisms; and the endosphere, the internal root tissues where endophytic microbes reside. Each niche harbours unique communities of eubacteria and methanogenic archaea, contributing to the functional diversity of the root microbiome. Tan et al. [70] explored the diversity and functional dynamics of the rice microbiome, encompassing a wide range of beneficial microbes, both within the plant and in the rhizosphere, that contribute to nutrient provision, act as natural biocontrol agents, and enhance plant growth while reducing dependency on agrochemicals. Islam et al. [71] showed that the rice microbiome exhibits remarkable diversity and functional dynamics, with microbes playing vital roles in key biogeochemical processes such as methane regulation, carbon sequestration, phosphate solubilization, and sulphur metabolism, all of which are essential for supporting rice growth and productivity. Adachi et al. [72] the root-associated microbiome of rice, showing that it varies significantly with soil fertilization status. In non-fertilized fields, nitrogen-fixing bacteria such as *Telmatospirillum* and *Bradyrhizobium* are more prevalent, supporting the adaptation of rice to nutrient-deficient environments. The rice rhizosphere harbours diverse microbial communities that interact with AM fungi, significantly influencing plant responses [73]. These microorganisms can directly enhance mycorrhizal colonization and fungal propagule abundance [74]. Moreover, agricultural practices can indirectly modulate rice mycorrhizal responses by altering the composition and function of the soil microbial community [75]. Juliyanti et al. [76] analyzed the microbial composition of the rhizosphere and endosphere of rice roots, finding no significant differences between weedy and cultivated rice lines. The endosphere had less diversity, with major groups including Proteobacteria, Myxococcota, Chloroflexota, and Actinobacteria.

These microbial communities are affected by geographical location, soil source, host genotype, and cultivation practice. Both biotic and abiotic variables have an impact on the microbial communities in rice field soils [77]. These include soil texture, the particular rice cultivar planted, pesticide applications, temperature, precipitation, humidity, pH levels, and the anions and cations balance [78].

Many species of oomycetes, archaea, and other microbes are crucial to the operation of many ecosystem services and ecological processes that support soil fertility and the health of rice [79]. In these communities, bacteria and fungi hold prominent positions despite their diversity. Ascomycota, Basidiomycota, and Glomeromycota are the most common members of fungal communities, whereas Proteobacteria, Chloroflexota, Actinobacteria, Bacteroidota, and Acidobacteria usually dominate bacterial communities. Understanding this intricate microbial diversity is essential for effective soil management and sustainable agricultural practices in rice cultivation. Coupling recently developed shot-gun metagenomics with bioinformatics tools offers enormous opportunities for shedding light on population diversity and functional aspects of the rice microbiome and their interactions with the host plants.

The microbiota associated with rice plants encompasses epiphytes on their surface, endophytes within plant tissues, and communities residing in the rhizosphere, endosphere, and phyllosphere (Figure 1). Rice plant-associated microorganisms can be mutualists, commensals, or pathogens, with pathogenic ones being of concern due to their economic impact [80]. The composition of the plant-associated microbiome is influenced by complex interactions between the host plant, microbial communities, and environmental factors. Within these communities, Proteobacteria and Firmicutes are typically enriched in plant endophytic environments, whereas Acidobacteria, Planctomycetes, Chloroflexota, and Verrucomicrobia are less abundant or depleted [49]. Ascomycota and Basidiomycota are the dominant fungal phyla in both above- and below-ground plant tissues. The phyllosphere harbours a diverse array of microorganisms, encompassing bacteria, fungi, actinomycetes, cyanobacteria, and viruses [81].

## 3. The Role of Rhizosphere Microbes in Rice Health and Growth

The rhizosphere refers to the soil zone within the 1–10 mm surrounding the plant’s roots, where the plant exerts influence through the release of mucilage, root exudates, and deceased plant cells [82]. The intricate interactions among plant roots, the rhizosphere microbiome, and abiotic factors create a dynamic and multifaceted structure commonly referred to as the rhizosphere complex. Under standard crop management practices, the rice rhizosphere microbiome is dominated by bacterial genera such as *Arenimonas*, *Arthrobacter*, *Anaeromyxobacter*, *Bacillus*, *Bellilinea*, and *Prevotella*. These microbes, along with mycorrhizal fungi, PGPR, and biocontrol agents, play a crucial role in enhancing the growth and disease resistance of rice plants [74]. In addition, AMF like *Glomus mosseae* can enhance rice plant height by 34%, shoot biomass by 122%, root biomass by 590% at the maximum tillering stage, and grain yield by 9.7% compared to non-inoculated plants [83] by improving Pi nutrition [84]. Approximately one million unique bacterial genomes could be present in only one gram of soil. Guo et al. [55] found that the rice rhizosphere microbiome exhibited an elevated α-diversity and a reduced β-diversity due to the homeostatic influence of the roots. Rice bacterial leaf blight (BLB) is a destructive phyllosphere bacterial disease caused by *Xanthomonas oryzae* pv. *oryzae* (*Xoo*) which significantly alters the bacterial–fungal community in the rhizosphere, decreasing bacterial diversity but not fungal diversity. Bacterial leaf blight significantly impacts the rhizosphere microbiome’s functional adaptation, increasing the abundance of functional genes involved in carbon, phosphorus, and methane metabolism [85]. Li et al. [86] found that inoculation with the R3 strain of *Herbaspirillum* altered the structure of native nitrogen-fixing microbial communities in the rhizosphere and endorhizosphere, increasing diversity and enhancing the abundance of key nitrogen-fixing genera (e.g., *Ralstonia*, *Azotobacter*, *Geobacter*, *Streptomyces*, and *Pseudomonas*). Additionally, the upregulation of nitrogen absorption and transport-related genes (*OsNRT1* and *OsPTR9*) in rice roots may contribute to increased yield. Pang and his colleagues isolated bacteria and fungi from upland rice roots in Xishuangbanna, China, in 2020 and they found that the root microbes differed from those within irrigated rice roots, with Firmicutes phylum members being enriched by 28.54% [87]. Their study also found that fungi from upland rice roots can increase plant growth under irrigated and drought-stress conditions, making them effective microbial resources for sustainable agricultural production in arid regions. Zhang et al. [88] showed significant differences in the microbiomes between wild and cultivated rice, with indica rice having more similar bacterial and fungal communities. Indica rice had the lowest proportion of Actinobacteria and the highest relative abundance of *Nitrospira*. However, the cultivated rice varieties (indica and japonica) had higher relative abundances of Magnaporthales and Ustilaginales.

Different inter-cultural operations like herbicide application also significantly reduced the abundance and diversity of soil microbial communities [89]. Indigenous bacterial strains under the genera *Burkholderia* and *Acinetobacter* significantly enhanced growth and yield parameters in rice under greenhouse conditions [90]. They stated that the persistence of certain taxa, including *Nitrosocosmicus*, *Nitrososphaera* (archaea), *Bacillus*, *Methyloceanibacter*, *Nitrospira*, *Thaurea* (bacteria), and *Acrophialophora*, *Aspergillus*, *Clonostachys*, *Emericellopsis*, *Exserohilum*, *Fusarium*, *Humicola*, *Nigrospora*, and *Pyrenochaetopsis* (fungi), improve the nutrient cycling and maintenance of soil fertility in rice agroecosystems. A study conducted by Hester et al. [91] found that the rice root microbiota is more sensitive to water management changes compared to bulk soil, with aerobic, plant growth-promoting bacteria (PGPB) being enriched under alternate wetting and drying (AWD) treatment. In contrast, anaerobic microorganisms were depleted, highlighting the potential of AWD to influence soil and root microbial communities differently.

Despite extensive studies on the impact of numerous isolated fungi and bacteria on rice growth and fitness, the intricate interplay between the root and rhizosphere microbiomes, and their collective role in rice growth, yield, and tolerance to various stresses remains poorly understood. As certain microorganisms have been demonstrated to regulate host gene expression, it is imperative to explore the far-reaching implications of rhizosphere microbiome structures. This investigation is essential for utilizing them in a bio-rational manner to sustainably enhance rice production while maintaining soil health.

## 4. Phyllospheric Microbes and Their Contributions to Rice Growth and Disease Resistance

The above-ground parts of rice plants, particularly their leaves, stems and nodes, provide a habitat that harbours a diverse range of microorganisms, many of which are essential for the plant’s growth and health [57]. Unlike the nutrient-rich rhizosphere and endosphere, the phyllosphere, which encompasses the area around or on the leaves, tends to be relatively low in nutrients [78]. It is a dynamic environment where resident microbes encounter fluctuating environmental factors such as temperature, water availability, humidity, solar radiation, and altitude.

Wang et al. [92] investigated the phyllosphere microbial communities of rice plant species at various elevations and growth stages. Their findings revealed a predominance of Proteobacteria, Actinobacteria, and Bacteroidetes among bacteria, and Ascomycota and Basidiomycota among fungi. These phyla exhibited significant variations across different elevations and growth stages. Elevation exerted a more pronounced influence on the α diversity of phyllosphere bacteria compared to fungi, while the growth stage significantly affected the α diversity of both bacterial and fungal communities. Additionally, the study demonstrated that the composition of both bacterial and fungal communities varied substantially along the elevation gradient within different growth stages.

The exploitation of rare earth minerals alters bacterial community structures in rice and reduces the diversity of dominant phyllosphere bacteria, such as *Burkholderia*, *Bacillus*, *Buttiauxella*, *Acinetobacter*, and *Bradyrhizobium*. However, it promotes bacteria that degrade pollutants and enhance nutrient availability, thereby supporting rice growth [82]. Phyla Actinobacteria and Firmicutes are often found in arid environmental conditions. Methylotrophic bacteria, primarily from the genera *Hyphomicrobium*, *Methylobacterium*, *Methylibium*, *Methylophilus*, *Methylocapsa*, *Methylocella*, and *Methylocystis*, are the dominant organisms found in the phyllosphere. Qi et al. [93] identified *Aspergillus* as the most prevalent fungal genus in rice grains. Notably, the mycotoxin-producing species *A. flavus* and *A. niger* were significantly more abundant. Yeo et al. [94] recorded that the prokaryotic community in rice landraces showed a high homogeneity across different landraces, with Proteobacteria being the most abundant phylum and an undefined genus of Cyanobacteria dominating. They found that rice plant organs, such as leaf blades and stems, primarily determine the composition of the prokaryotic community.

Roman-Reyna et al. [95] have identified 12 genera that characterize the microbial community on rice leaves, ranging from commensal or pathogenic to beneficial, including the phyla of Proteobacteria, Firmicutes, Actinobacteria, Cyanobacteria, Tenericutes, and Euryarchaeota. Proteobacteria appear to be the predominant colonizers, with Bacteroidetes and Actinobacteria also being prevalent in phyllospheres. Both biological and environmental influences, along with human activities such as farming methods and fertilizer use, play essential roles in shaping both the taxonomical and functional aspects of phyllosphere microbiomes. These diverse factors contribute to the dynamic and heterogeneous nature of the rice phyllosphere environment.

## 5. Endophytic Microbes and Their Role in Enhancing Stress Tolerance in Rice

Certain microorganisms, known as endophytes, have the ability to penetrate and occupy the internal tissues of plants, giving rise to the endospheric microbiome. This pertains to the internal realm of plants, where endophytes, encompassing both bacteria, fungi and viruses, within plant tissues reside for most of their life cycle without inducing pathogenic symptoms [96]. Beneficial microbes enhance nutrient acquisition by solubilizing essential nutrients like nitrogen and phosphorus, producing phytohormones such as indole-3-acetic acid (IAA), auxins, and cytokinins to regulate root and shoot development, while synthesizing enzymes, secondary metabolites, and siderophores that sequester iron, limiting its availability to pathogenic fungi and thereby protecting plants from diseases [97,98,99,100]. Microbes bolster plant resilience to abiotic stresses like drought and salinity by activating antioxidant systems, promoting osmolyte accumulation, improving root architecture for better water and nutrient absorption, and protecting rice from diseases through antimicrobial compound production or inducing systemic resistance without triggering harmful immune responses, thus maintaining a symbiotic balance with the host plant while promoting overall growth and stress tolerance [101].

Tian et al. [102] identified 96 endophytic bacterial strains with plant growth-promoting (PGP) traits from *Oryza officinalis*, with 11 strains (*Enterobacter mori*, *E. ludwigii*, *E. cloacae*, *Bacillus amyloliquefaciens*, *B. siamensis*, *Pseudomonas rhodesiae*, and *Kosakonia oryzae*) demonstrating enhanced root development, biomass accumulation, chlorophyll content, and nitrogen uptake in perennial rice seedlings. Wu et al. [103] demonstrated that, under abiotic stress (e.g., drought stress), rice may specifically enrich certain bacterial taxa (e.g., Actinobacteriota, Gemmatimonadetes, Patescibacteria, Bacteroidetes and Firmicutes). These bacteria establish a positive interaction with the host rice root system and help to improve adaptation. They are a good indicator of stress tolerance under stressful conditions. Wang et al. [57] found that other microorganisms like *Xanthomonas sacchari* JR3-14 enhance the diversity and complexity of rice-associated bacterial communities, particularly in the root and stem endosphere during the early growth stages. They also found that the assembly of other bacterial communities like *Bacillus*, *Azospira*, *Azospirillum*, and *Arthrobacter*, can increase the number of seed endophytes. These bacterial communities help in crop production and breeding programmes. Zhang et al. [104] revealed that rice microbiome assemblages are primarily shaped by the microhabitat rather than geographic location or rice variety, with diversity and complexity decreasing from soil to seed. Notably, core bacterial endophytes, especially *Pantoea* and *Xanthomonas*, were vertically transmitted across generations, displaying plant growth-promoting traits but limited antagonism to major rice pathogens.

The AM fungi and other endophytic fungi are the primary inhabitants of the endosphere, increasing nutritional acquisition and improving the stress resilience of the host plants [105,106]. During the vegetative phase of rice plants, bacteria capable of producing siderophores, notably from the genera *Sphingomonas*, *Pseudomonas*, *Burkholderia*, and *Enterobacter*, are predominantly found in plant tissues. These bacteria play a vital role in facilitating the plant’s acquisition of iron and other essential elements from the soil. Besides colonizing plant organs, endophytic bacteria are also associated with rice seeds. Genera belonging to the *Alphaproteobacteria*, *Gammaproteobacteria*, *Flavobacteria*, *Bacilli*, and *Actinobacteria* classes have been observed to inhabit rice seeds. Some of these endophytic microorganisms engage in beneficial interactions with their host plants. Endophytic microorganisms contribute to enhancing plant health, performance, and adaptation to both biotic and abiotic stresses [107]. Moreover, the composition and diversity of the endophytic microbiome within above-ground and below-ground tissues may vary within the same plant. By understanding the mechanisms underlying beneficial plant–microbe interactions, researchers can develop novel crop improvement strategies, such as using endophytic bacteria as biofertilizers and biocontrol agents, to reduce reliance on chemical inputs and create more resilient and productive crop varieties.

## 6. Decoding Signalling Pathways in the Rice Rhizosphere

The signalling mechanisms in the rhizosphere can be classified into three main types: (i) Plants secrete low-molecular-weight compounds to communicate with microbes, fostering complex interactions essential for nutrient acquisition and stress resilience. (ii) Interspecies and intraspecies microbial signalling primarily occur through quorum-sensing (QS) and other signalling processes. This allows microorganisms to coordinate and adjust their behaviour based on population density and their functional behaviour. (iii) Microorganisms inherent to plant signalling involve compounds produced by microorganisms that influence various aspects of plant physiology and immunity that ultimately impact root system architecture, plant defence mechanisms, responses to both biotic and abiotic stresses, and gene expression [108]. Acyl-homoserine lactones (AHLs) regulate quorum sensing, enabling bacteria to coordinate behaviours like biofilm formation, antibiotic production, and root growth modulation, which enhances root elongation, lateral root formation, and root hair density [109]. Conversely, volatile organic compounds (VOCs) act as chemical signals between microbes and plants, priming the plant’s immune responses, inducing systemic resistance to pathogens, and influencing nutrient cycling processes like nitrogen fixation and phosphorus solubilization by attracting beneficial microorganisms to the rhizosphere, which aid in nutrient uptake [110,111].

For instance, several researchers have investigated the role of AHLs and other signalling molecules in shaping plant–microbe interactions. Updated research highlights how these molecules not only regulate QS among microbial populations but also influence plant physiological responses, such as stress tolerance and nutrient uptake. Chen et al. [112] found that AHL-mediated signalling in the rice rhizosphere promotes the recruitment of beneficial microbes, and enhances nitrogen fixation and root growth. Similarly, Zhang et al. [63] demonstrated that certain AHLs induce systemic responses, improving the plant’s defence against pathogens.

Additionally, beyond AHLs, other signalling molecules like VOCs and lipo-chitooligosaccharides (LCOs) have been shown to play critical roles in plant–microbe signalling. Wang et al. [92] uncovered that VOCs released by *Pseudomonas* spp. in the rice rhizosphere can alter root architectures, improving water and nutrient acquisition. Kelbessa et al. [113] expanded the role of LCOs in promoting symbiotic relationships, particularly in phosphate-solubilizing bacteria, which significantly enhances nutrient cycling in paddy fields. These recent publications underscore the complex and multi-faceted nature of microbial signalling in rice ecosystems. Integrating these studies would provide a richer, more nuanced discussion of how microbial communication contributes to plant health, nutrient efficiency, and resilience. Addressing these aspects would not only align better with the objectives of this review but also offer a more up-to-date and robust understanding of the microbial plant signalling networks that are crucial in rice production systems.

Santosh Kumar et al. [114] reported that rice seedling roots interact with *Sinorhizobium meliloti* 1021, which secretes bioactive signalling chemicals. Such chemicals are recognized by rice receptor proteins (PRRs, FLS2, and LRR-RLKs), leading to changes in gene expression in the plant. These bioactive molecules interact with rice root cells close to where the bacteria colonize and initially infect the plant. Once these bacterial signals are identified and transmitted, the differentially expressed genes (DEGs) associated with producing phytohormones that control plant growth and development, like auxins, gibberellins, and cytokinins, change in the rice seedling shoots. Subsequently, the bacteria migrate within the plant to other rice tissues where they coexist symbiotically, continuing to influence plant gene expression.

Xiao et al. [115] reported that the combined application of *Rhodopseudomonas palustris* and *Bacillus subtilis* resulted in a substantial increase in rice yields, reaching up to 17.73%. This synergistic effect was attributed to the significant modification of the soil bacterial community structure. Furthermore, the study demonstrated an enhancement of membrane transporters and signal transduction pathways, coupled with an increase in certain essential metabolic pathways.

AHLs are the primary signalling molecules involved in quorum sensing QS and are typically species-specific [63]. These signaling molecules function by interacting with bacterial receptors, leading to alterations in gene expression. This enables different microbial populations to synchronize their activities, thereby allowing them to function as a unified and coordinated community [116]. Viswanath et al. [116] demonstrated that QS systems using AHLs are integral to communication among bacterial species within the rhizosphere. They highlighted how AHL-mediated QS can influence the interactions between microbes, affecting traits like biofilm formation, motility, and the production of secondary metabolites. These interactions are vital for establishing a balanced microbial community that can adapt to environmental changes and compete with other microbes in the soil environment.

QS plays a crucial role in structuring the rhizosphere microbiome, enabling the formation of microbial consortia that can have either beneficial effects, such as promoting plant growth and resilience, or pathogenic impacts [5]. Recent findings have reported the production of AHLs in the rice rhizosphere by species such as *Acinetobacter lactucae*, *Aeromonas popofi*, *Serratia oryzae*, and *Rhizobium wuzhouense* for the first time. The presence of diverse AHLs among these rhizobacterial groups suggests that these signalling molecules might be key regulators in coordinating rhizobacterial behaviour and promoting symbiotic interactions between microbes and plants. This discovery sheds light on the potential role of AHLs in regulating the dynamic interactions within the rice rhizosphere, contributing to enhanced plant health and growth. The establishment of a beneficial microbial community in the rice rhizosphere relies on signalling events between the host plant and bacterial symbionts. Molecular signals, including the well-known Nod factor, facilitate the selective recruitment of specific microbes by the plant. This process involves the exchange of signals, such as flavonoids inducing bacterial Nod factor production. Tricarboxylic acid secretion also contributes to recruiting other beneficial microbes, enhancing overall plant health. Cooperative interactions among different microbes shape the plant’s well-being. It is a common occurrence that VOCs produced by microorganisms can also stimulate both plant growth and tolerance to biotic stresses.

Plants can detect and respond to various VOC signals originating from PGPR or plant growth-promoting fungi (PGPF), including compounds like undecanone and heptanol. VOCs such as 2-heptanol and 2-undecanone, released by *Bacillus subtilis* and *B. amyloliquefaciens*, have been identified to enhance the growth performances of *Arabidopsis thaliana* when grown in the presence of these PGPR strains. Therefore, microbially produced VOCs influence plant physiology, foster growth, and fortify resistance against biotic stress [117]. VOCs also induce the systemic resistance of rice plants to biotic and abiotic stresses.

Plants activate signalling pathways to defend against various threats, such as insects and pathogens. Different pathways, along with jasmonic acid (JA)/Ethylene (ET)-dependent Induced Systemic Resistance (ISR) and salicylic acid (SA)-dependent SAR, are associated with plant defence responses [118,119]. Herbivore feeding habits induce JA-SA signalling, activating defence responses in distal organs. Aboveground herbivory can recruit rhizobacteria and induce systemic signalling throughout the plant. Plant immune signalling influences microbial communities in the root microbiome, with changes in JA signalling impacting the rhizosphere community. Signalling compounds, such as hormones (JA, ET, SA, ABA, CK, GA, auxin), are essential. Plants release root exudates in response to changes in JA signalling, influencing bacterial and archaeal abundances in the rhizosphere. The production of signalling compounds is triggered by stimuli like herbivory, pathogen attack, and environmental stress. Signalling pathways involve specific receptors, initiating cascades leading to the synthesis and release of compounds.

Rice root exudates indeed consist of a diverse array of primary and secondary metabolites, encompassing sugars, amino acids, organic acids, phenolics, and flavonoids [120,121]. The composition and stoichiometric ratios of these exudates have been shown to impact microbial activity, carbon mineralization, and microbial biomass in the rhizosphere, thereby modulating the biotic interactions and nutrient cycling in the soil ecosystem. Some beneficial endophytes like *Azoarcus* activate JA and SA signalling, promoting root colonization and shaping the root microbiome of rice [122]. Endophytic fungus *Acrocalymma vagum* enhanced rice yield by 5.73% and induced 83.24% resistance against rice blast disease. Zeng et al. [123] found that some *Lactobacillus* and *Nigrospora* strains show significantly inhibitive activity against the rice blast pathogen *Magnaporthe oryzae*. They also found that some fungal strains of *Sarocladium* and *Nigrospora* genera promoted rice growth. This allows plants to communicate with microbes and regulate various physiological processes, including defence responses and growth. The bacterial biosynthesis of phytohormones offers a promising approach to enhance plant development.

## 7. How Microbes Enhance Rice Growth and Yield: Mechanisms and Benefits

Microbes play diverse roles in influencing the morphology, physiology, growth, resilience, and gene expression patterns of plants [79]. The application of microorganisms in rice production is regarded as a sustainable and environment-friendly approach that has gained attention in agricultural practices without the use of agrochemicals or mechanical interventions. PGPRs enhance rice yield through various mechanisms, including the production of phytohormones like auxins and cytokinins that regulate plant growth, promote root formation, and improve overall plant health [124,125], solubilizing phosphorus for better nutrient uptake, which is crucial for root development and grain formation [126] increasing iron availability through siderophore production to limit pathogen growth in iron-deficient conditions [127,128], and suppressing soil-borne diseases by producing antimicrobial compounds that reduce the impact of pathogens like rice blast and bacterial blight [129,130]. These types of functionalities absolutely highlights the importance of beneficial bacteria in promoting plant growth through various mechanisms beyond nitrogen fixation. While nitrogen-fixing bacteria (for instance *Rhizobium*) are traditionally connected with leguminous plants, other groups of bacteria, like PGPR, can positively influence the growth and yield of non-leguminous crops, including rice. Biological nitrogen fixation with *Rhizobium* spp. can significantly increase shoot and root growth, grain yield, biomass, nutrient uptake, and nitrogen use efficiency in rice plants by activating the conversion of atmospheric nitrogen (N_2_) into ammonia (NH_3_) via the nitrogenase enzyme, which plants can readily assimilate, enhancing their nitrogen availability [131]. Rice yields improved by 17.73% with the combined use of *Rhodopseudomonas palustris* and *Bacillus subtilis* due to changes in the soil bacterial community [114]. Microbial additives decreased greenhouse gas emissions and dry matter losses by improving the fermentation process through enhanced nutrient preservation, optimized microbial networks, and sustainable crop residue utilization [132]. Kalkhajeh et al. [133] demonstrated that combining basal nitrogen fertilizer and straw-decomposing microbial inoculant (SDMI) significantly enhances wheat straw decomposition and rice yield as its application boosted microbial activity and respiration, particularly in the tillering stage, leading to higher straw decomposition and biomass production.

Co-inoculation with nitrogen-fixing bacteria and PGPB, such as *Rhodopseudomonas palustris* and *Bacillus subtilis*, resulted in yield increases of up to 13.7% due to improved seed setting rates and altered microbial community structures [134]. The use of biofertilizers allows for a reduction in nitrogen fertilizer application while maintaining or even increasing rice yields by up to 26%, thus mitigating nitrogen losses and environmental impacts [135]. Methane-derived microbial biostimulants have been found to increase rice grain yield while simultaneously reducing greenhouse gas emissions, demonstrating a dual benefit for food security and environmental sustainability [136]. Different crop establishment methods combined with microbial formulations showed improved soil microbial properties and increased rice yields, particularly with specific combinations of nutrients and microbial inoculants [137]. The use of consortia of PGPB (*P. agglomerans* strain O4, *P. putida* strain P13 + *P*. *agglomerans* strain P5, *P. koreensis* strain S14 + *P*. *vancouverensis* strain S19) significantly enhanced rice grain yield, protein content, and nutrient uptake compared to untreated plants [138]. Coinoculation with *Azospirillum brasilense* and *Pseudomonas fluorescens* significantly increases rice yields while reducing mineral nitrogen use (30 kg of N ha^−1^), enhancing nitrogen efficiency and economic production (37%) [139]. Inoculating rice with effective microbial agents increases crop performance by mobilizing nutrients, producing phytohormones, and providing protection against biotic and abiotic stresses, ultimately improving yield [140]. The combination of *Trichoderma* microbial inoculant and rice straw compost significantly increased rice yield while reducing soil copper and lead concentrations in contaminated rice [141].

## 8. The Role of Microbes in Alleviating Biotic Stresses in Rice

The metaorganism concept, which encompasses plants and their associated microorganisms, has proven valuable in enhancing agricultural practices and disease management. In rice cultivation, this concept has been instrumental in promoting biotic stress tolerance and reducing the need for chemical pesticides, which, although effective, pose risks to the environment and human health [142]. Beneficial microbes in the rice rhizosphere establish mutualistic relationships that support plant growth and resilience against pathogens through mechanisms like competition, antibiosis, and the production of extracellular enzymes [143]. As biocontrol agents, these microbes mitigate plant diseases responsible for up to 25% of global crop losses, particularly in developing regions [144]. Their interactions with rice plants facilitate adaptation to biotic stressors and enhance tolerance, providing a sustainable alternative to chemical interventions in agriculture [5].

Beneficial microorganisms, such as bacteria, fungi, and other microbes, play a pivotal role in safeguarding plants from pests, diseases, and pathogens. These microbes exert antagonistic and biocontrol effects, thereby enhancing plant defence mechanisms. Their interactions with plants, both direct and indirect, contribute significantly to the plant’s resilience against infections and infestations. For instance, many researchers, like the authors of [145,146,147,148,149,150,151,152], have proven the efficacy of several strains of *Bacillus subtilis* in controlling the most devastating blast disease affecting rice caused by rice *Magnaporthe oryzae*. This bacterium produces antibiotics like surfactins and iturins that suppress the growth of blast fungal pathogens. In addition, they exhibited the ability to synthesize cell wall-degrading enzymes such as chitinase, protease, and β-1,3-glucanases, as well as antifungal metabolites such as siderophores. Known for inducing systemic resistance in rice, the bacterium *Pseudomonas fluorescens* can help to manage bacterial blight caused by *Xanthomonas oryzae* pv. *oryzae* (*Xoo)*. *P. fluorescens* induces systemic resistance in rice by activating defence pathways through jasmonic acid and ethylene signalling, while producing antibiotics like 2,4-diacetylphloroglucinol (DAPG) and lytic enzymes that directly inhibit *Xoo*. Additionally, it promotes plant growth through phytohormone production and competes with pathogens for resources in the rhizosphere by producing siderophores and forming biofilms [153,154]. The beneficial fungus *T. harzianum* is used in rice fields for the management of *Rhizoctonia solani*, the causal agent of sheath blight in rice. *T. harzianum* acts through direct parasitism of the pathogen by secreting cell wall-degrading hydrolytic enzymes, and it also competes with the pathogen for nutrients. Several strains of *Trichoderma* have been proven to be effective against rice blast [155,156,157] and root rot caused by *Fusarium* spp. [158,159], brown leaf spot disease caused by *Bipolaris oryzae* [160], rice sheath blight disease caused by *Rhizoctonia solani* [161], and sheath rot caused by *S. oryzae* [162]. Furthermore, the *T. asperellum* isolate demonstrated a pronounced ability to inhibit the growth of *R. solani* by means of hyphal coiling [163]. The nematophagous fungus *Paecilomyces* spp. has shown efficacy in controlling rice root knot nematodes (*Meloidogyne graminicola*) by parasitizing the nematode eggs and reducing their population. *Pochonia chlamydosporia* also works similar way [164]. PGPR like *Azospirillum brasilense* are not only used to enhance rice growth but also to induce systemic resistance against diseases like rice blast and bacterial blight by priming the plant’s immune system [165]. PGPR, such as *Bacillus velezensis*, *B. megaterium*, and *B. toyonensis*, significantly improve germination, seedling vigour, and dry weight. They have also demonstrated the lowest disease incidence, relative lesion length, and delayed sclerotia formation, and the maximum grain yield was recorded by suppressing pathogens through mechanisms like nutrient competition, the production of antimicrobial compounds, and the induction of systemic resistance [166]. A *Bradyrhizobium japonicum* strain capable of abscisic acid biosynthesis demonstrates potential as a biological control agent for bacterial wilt disease induced by *Ralstonia solanacearum* [166].

These biotic stresses, caused by the aforementioned devastating pathogens, lead to significant yield losses in rice production systems. Table 1 highlights the role of antagonistic microbes in suppressing phytopathogens, reducing diseases, promoting plant health, and enhancing resilience, offering sustainable and eco-friendly strategies for managing biotic stress challenges in rice cultivation.

## 9. Harnessing Microbes to Combat Abiotic Stresses in Rice

Rice is severely susceptible to abiotic stresses like salinity, drought, heat, cold, acidity, and sodicity, as well as biotic stresses caused by insects and pathogens (Figure 2). Plants experiencing stress can leverage beneficial microbial endophytes for survival. By emitting chemical signals, plants essentially “cry for help”, activating these microbes to mitigate stress-related damage [194]. Microbes like *Bacillus*, *Pseudomonas*, and *Trichoderma* can alleviate these abiotic stresses by producing antioxidants, enzymes, antibiotics, and phytohormones (Table 2). Microbial involvement in rice plants is multifaceted, contributing to stress tolerance through various mechanisms. Whether through direct interactions with pathogens, modulation of plant physiology, or improvements in soil conditions, the plant–microbe partnership plays an essential role in enhancing the resilience of rice crops to both biotic and abiotic stressors [195]. Microbes produce osmoprotectants, such as proline and trehalose, which help rice plants maintain cellular integrity under stress by stabilizing proteins and cell membranes [196]. They also enhance antioxidant defence systems, promote root growth, improve water retention, and induce systemic resistance by activating stress-responsive genes, all of which contribute to better resilience against abiotic stresses like salinity and drought [197,198,199].

Microbes from the genera *Bacillus*, *Pseudomonas*, *Enterobacter*, *Ochrobactrum*, *Alcaligens*, *Paecilomyces*, *Burkholderia*, *Achromobacter*, *Azospirillum*, and *Glomus* have been shown to mitigate abiotic stress in rice. These beneficial microorganisms achieve this by enhancing the production of antioxidants, hormones (ascorbate, proline, ethylene, and auxin), and stomatal conductance, as well as by synthesizing compounds like 1-aminocyclopropane-1-carboxylate deaminase, β-aminobutyric acid, salicylic acid, and siderophores. This collective action significantly contributes to the plant’s resilience against environmental challenges [31].

Different microbial agents (endophytic and rhizospheric bacteria) exhibit varying mechanisms of salinity stress alleviation in rice, potentially enhancing photosynthetic efficiency, root architecture, and antioxidant enzyme activity viz. CAT, SOD, PO, PPO, APX, and PAL activity, along with an effect on proline levels, and also inhibits the activities of superoxide dismutase and lipid peroxidation. *Bacillus haynesii* 2P2 demonstrated the most significant improvement in biomass accumulation and tiller number, indicating a possible cultivar-specific microbial consortium for climate-resilient rice cultivation by stimulating stress-responsive genes, i.e., *CATa*, *cAPX*, *MnSOD*1 [222]. Furthermore, the rice plant’s tolerance to salt was enhanced, and its capacity to generate glycine betaine was elevated. *Bacillus subtilis* and *B. pumilus* enhance plant growth in saline environments through their ability to solubilize phosphate and secrete the phytochemicals hydrogen cyanide (HCN), indole-3-acetic acid (IAA), and ammonia. *Bacillus amyloliquefaciens* is known to induce salt tolerance in rice plants by stimulating the production of phytohormones, including auxins and abscisic acid (ABA) [223]. Shahzad et al. [224] found that nitrogenase activity and IAA were significantly increased by the application of PGPR like *Azospirillum amazoense*.

In addition to this, *Stenotrophomonas maltophilia* application increased ACC deaminase, and phosphate solubilization in rice [225]. Both of these PGPR improve stress tolerance in rice plants. *Curtobacterium* sp., *Enterobacter ludwigii*, *Bacillus cereus*, and *Micrococcus yunnanensis* are useful for mitigating salinity stress in rice. Several investigations have demonstrated the beneficial impact of ACC deaminase-producing microorganisms, like *B. pumilus* strain TUAT-1, on plant growth in saline conditions, as exemplified by their positive influence on rice [183]. *Pseudomonas* sp. inoculation enhances rice plants’ tolerance to salt stress [226], while *B. pumilus* inoculation mitigates the adverse effects of both salt and high boron stress on rice plants [207].

Members of the *Trichoderma* and *Pseudomonas* genera mitigate the adverse effects of water scarcity on rice plants. Microbial inoculants enhance the accumulation of polyphenolic compounds, known for their antioxidant properties. The activation of superoxide dismutase, coupled with the accumulation of hydrogen peroxide, was associated with hypersensitive cell death in leaves. Microbial inoculation also upregulates the activities of peroxidase, ascorbate peroxidase, glutathione peroxidase, and glutathione reductase enzymes, contributing to a reduction in the reactive oxygen species (ROS) burden. Genes involved in key metabolic pathways, including phenylpropanoid biosynthesis, superoxide dismutation, hydrogen peroxide peroxidation, and oxidative defence, were found to be overexpressed. These findings demonstrate the efficacy of microbial inoculants in bolstering the intrinsic biochemical and molecular capacities of rice plants to cope with drought stress [227]. Foliar spraying of *B. megaterium* PB50 produces IAA which induces the synthesis of ACC deaminase in the rice plants, which enhances drought stress tolerance in the rice plants [228]. ACC deaminase-producing rhizobacteria have demonstrated their efficacy in mitigating drought stress in rice. An overexpression of the trehalose-6-phosphate synthase gene *OsTPS1* in rice plants, using *Escherichia coli* or *Saccharomyces cerevisiae*, significantly enhanced their drought tolerance. The presence of *Patescibacteria* and the genera *Massilia*, *Nocardioides*, *Frateuria*, and *Angustibacter*, along with fungi belonging to *Talaromyces*, may enhance rice drought tolerance [229].

Chieb and Gachomo, [230] stated that an increased proline content in rice plants, due to inoculation with PGPR consortia, enhances the plants’ tolerance to water stress. The application of *Pseudomonas* sp. K32 improves the Pb and Cd stress tolerance in rice. *Pseudomonas* species demonstrate the ability to remediate heavy metal contamination by detoxifying cadmium citrate and the Fe (III)–zinc complex through biotransformation processes. Saha et al. [231] reported that bacteria-produced siderophores can chelate not only ferric ions but also other metals, thereby assisting in phytoremediation. Consequently, PGPR can mitigate the adverse effects of heavy metal stress on plants. Inoculation with the *Azotobacter brasilense* strain A39 facilitates the accumulation of polyamines in rice seedlings subjected to osmotic stress, as demonstrated by [232]. Psychrophilic bacteria, including *Arthrobacter nicotianae*, *Brevundimonas terrae*, and *Pseudomonas cedrina*, have demonstrated their efficacy to enhance plant growth and development even in adverse, and frigid environments. Extensive research has been undertaken to identify bacteria capable of producing organic acids that can solubilize insoluble phosphate [233]. In environments characterized by elevated temperatures, the bacterium *Paecilomyces formosus* exhibited traits that positively influenced plant growth. Similar to their bacterial counterparts, endophytic and symbiotic fungi, including *Curvularia protuberata*, demonstrated the capacity to bolster plant heat tolerance and positively impact plant development.

In response to salt stress, *T. harzianum* increases rice plants’ resistance by improving physiological parameters under saline conditions [195]. *B. pumilus* enhances tolerance to salt stress in rice by upregulating antioxidant enzymatic activities and accelerating certain soil enzyme activities [234]. *Piriformospora indica* regulates salt tolerance in rice by enhancing specific genes associated with salt stress responses [235]. The bacterial endophyte *P. alhagi* enhances antioxidant enzyme activities and overall plant performance in salt stress conditions [236].

## 10. Metagenomics: Unravelling the Complexities of Rice Microbial Communities

Metagenomics has emerged as a pivotal tool in understanding the intricate associations between microbes and plants, providing unprecedented insights into the complex dynamics of plant-associated microbial communities. Extracting DNA from plant tissues is challenging due to the dominance of plant DNA over microbial DNA. Plant-associated microbes, especially endophytes, are under-investigated sources of bioactive molecules [237]. Metagenomics studies can help uncover beneficial functions of endophytes and identify potential beneficial species. DNA extraction methods for plant-associated microbial communities include surface sterilization and the use of commercial kits. Metagenomic techniques allow for the analysis of endophyte communities and their functional roles in plants. These advanced techniques involve profiling the microbial community using genomic samples from the natural habitat [80]. Comparative studies utilizing metagenomics have allowed for the classification of microbes into different taxa based on gene libraries, highlighting significant differences among them [238]. In addition to metagenomic studies, other omics technologies, such as metabolomics and proteomics, have been employed to assess microbial diversity [239]. The success achieved through next-generation sequencing (NGS) in accessing metagenome sequences has been remarkable in various fields, including medicine, agriculture, pharmaceuticals, food, and environmental studies, facilitated by the use of different metagenomes. Metagenomic studies provide a comprehensive understanding of the taxonomic classification and functional/metabolic pathways of active endophytic microorganisms [50,240]. This approach is particularly valuable for determining the total microbiomes, including yet-to-be-cultured environmental microbes. Despite the computational challenges involved in the analysis and interpretation of metagenome sequenced data, the interest of scientists in metagenomics studies remains high.

The generation of longer sequence reads in metagenomics studies significantly influences the analysis of metagenomic sequences. Notably, real-time nanopore sequencing analysis has emerged in clinical metagenomic studies and has been applied in the detection of microbial pathogens for genomic surveillance [241]. This technology allows for the direct sequencing of DNA without the need for amplification and holds promise for enhancing our understanding of microbial communities in diverse environments.

Metagenomic analyses provide a sophisticated approach for elucidating the intricate microbial communities that engage with plants [241]. By leveraging the advantageous characteristics of beneficial microorganisms, this technology offers prospects for augmenting crop yield and robustness. High-throughput sequencing has become a prevalent tool for investigating the microbial communities associated with various crops and their interactions with environmental challenges [242,243]. Metagenomic binning offers a precise method for characterizing the taxonomic diversity and functional capabilities of microbial communities at a fine-grained level [244]. Numerous researchers [245,246,247,248,249] utilized metagenomic to identify and analyze the genetic material of all of the microorganisms present in a given environment. This helps in understanding the composition, functions, and interactions within the microbial communities. Cheng et al. [244] integrated metagenomics and machine learning to investigate the root-associated microbiome of two rice cultivars to identify clues for enhancing crop resilience to Cd stresses. Similarly, Li et al. [250] combined metagenomics with metabolomic analyses and found that Bt rice (T1C-1) planting increased soil microbiome diversity and network stability, and influenced carbon and nitrogen cycling without adversely affecting probiotic or phytopathogenic microorganisms, though significant differences were observed in the rhizosphere compared to non-Bt cultivars. Additionally, Bt rice selectively modulated rhizosphere microbiota through altered root exudates, impacting soil metabolite profiles and providing mechanistic insights into the plant–microbe–environment interactions of genetically modified crops. A proteomics analysis of rice where *Pseudomonas alcaliphila* Ej2 was used as a biocontrol for blast under salt stress revealed rice proteomic profile, including metabolism, plant–pathogen interactions, and the biosynthesis of unsaturated fatty acids significantly influenced by the applied bacterial strain [123]. The proteomic analysis of rice treated with *Pseudomonas alcaliphila* Ej2 as a biological control agent for blast disease under salt stress conditions identified significant alterations in rice protein expression profiles related to metabolic processes, plant-pathogen interactions, and the biosynthesis of unsaturated fatty acids. These findings underscore the substantial influence of the applied bacterial strain on rice physiology [123]. Transcriptomics analysis is also used to map the functions of rice plant associated microbiome. Patel et al. [251] conducted a transcriptomic analysis of the endophytic bacterium *Microbacterium testaceum* and found that it can inhibit the growth of the rice blast fungus *Magnaporthe oryzae* through the release of VOCs and the upregulation of the *OsNPR1* and *OsCERK* genes. A comparative analysis of microbiome composition in rice leaves using metabarcoding and culture-based methods identified bacterial communities (*Pantoea*, *Enterobacter*, *Pseudomonas*, and *Erwinia)* associated with blast disease resistance. A quantitative RT-PCR analysis revealed an elevated expression of defence-related genes, including *OsCEBiP* and *OsCERK1*, and phytohormone-associated genes like *OsPAD4*, *OsEDS1*, *OsPR1.1*, *OsNPR1*, *OsPDF2.2*, and *OsFMO*, in rice seedlings [252]. Nanfack et al. [52] utilized Illumina-based 16S rRNA gene sequencing to analyze the bacterial communities in two rainfed rice varieties, NERICA 3 and NERICA 8. Their findings revealed that healthy seedlings exhibited a higher abundance of beneficial genera, such as *Brevundimonas*, *Sphingomonas*, and *Exiguobacterium.* Conversely, abnormal seedlings displayed an increase in potentially harmful genera originating from the seed-associated microbiome. Krishnappa et al. [253] utilized 22 meta-barcoded NGS datasets to map the rice foliar microbiome across various plant surfaces, revealing a diverse microbiome of 157 genera, with Proteobacteria and Actinobacteria as the dominant phyla. In addition, their culturomics confirmed the prevalence of beneficial genera like *Pantoea* and *Pseudomonas*, highlighting potential microbial resources for microbiome-assisted rice cultivation. Sondo et al. [254] used similar techniques to analyze the diversity and plant growth promoting ability of rice root-associated bacteria in Burkina-Faso.

## 11. Microbiome Engineering: A Pathway to Sustainable Rice Cultivation

Plant microbiome engineering refers to the manipulation and modification of the microbial communities associated with plants [255]. Ongoing research in this field aims to optimize microbial interactions to benefit rice cultivation and contribute to global food security. There are different approaches to rice microbiome engineering, including biotechnological and conventional methods (Figure 3).

Liu et al. [256] outlined a protocol for CRISPR-Cas9 techniques, such as designing target sequences, constructing expression vectors, and transforming *Pyricularia oryzae (synonym: Magnaporthe oryzae*) for genome editing. These methods enable efficient gene disruption, base editing, and reporter gene knock-in without altering host components. Additionally, this protocol can facilitate the application of CRISPR-Cas technologies in various functional genomics studies of *P. oryzae*. The study conducted by Li et al. [257] demonstrated increased plant resistance against blast disease caused by *M. oryzae* using a targeted CRISPR-Cas9 mediated mutation in the ethylene responsive factor (ERF) *OsERF922* in rice. For example, a recent study by Zafar et al. [258] revealed that a solitary amino acid substitution in the ALS gene, specifically the replacement of tryptophan with leucine at position 548, significantly improved the resistance of basmati rice to bacterial blight. *OsSWEET14* gene editing by Sam et al. [259] improved BB resistance in rice. Mathsyaraja et al. [260] enhanced blast disease resistance in rice through CRISPR-Cas9-mediated editing of the *OsHDT701* gene. Similarly, Park et al. [261] successfully improved disease resistance by targeting the Mildew Locus O (*CaMLO2*) gene using CRISPR-Cas9 technology [262]. And other researchers successfully developed rice plants that were resistant to the bacterial blight pathogen *Xoo* by silencing the *OsSWEET11*, *OsSWEET13*, and *OsSWEET14* genes, which regulate sugar transport within the plant. Ji et al. [263] demonstrated that the function of the bacterial blight resistant gene *Xa23* could be restored by correcting gene mutations using the CRISPR-Cas9 system. This restored gene function triggered effective defence responses against *Xoo* infection. Engineering the endogenous *Xa23* gene by EBE stacking in its promoter broadens *Xa23*’s defensive role against diverse bacterial pathogens, addressing the short-lived resistance typically conferred by a single R gene, which pathogens often overcome through mutations within effector genes [264]. CRISPR-Cas9-mediated editing of *OsSWEET13*, a susceptibility gene for bacterial blight in rice caused by *X. oryzae* pv. *oryzae*, generated two knockout mutants targeting its promoter, leading to enhanced tolerance against bacterial blight in rice [265]. IR64 rice lines were engineered to exhibit resistance to rice tungro spherical virus (RTSV) through targeted mutagenesis of the eukaryotic initiation factor 4G (eIF4G) gene. CRISPR-Cas9-mediated mutagenesis generated novel eIF4G alleles, leading to the development of RTSV-resistant rice varieties [266]. Plant annexins play a noteworthy role in plant improvement and defence against various environmental stresses, as highlighted by Shen et al. [267] who demonstrated the significance of the rice annexin gene *OsAnn3* under cold stress through studies on *OsAnn3* CRISPR knockouts. Shen et al. [268] employed CRISPR-based Quantitative trait loci (QTL) editing to elucidate the roles of grain number (Gn1a) and grain size (GS3) QTLs in rice varieties. Shen et al. [267] previously engineered 571 *OsALS* genes and identified three mutant variants—ALSS627N and 1884G-A, ALSS627N, and ALSS627N/G628E—that demonstrated resistance to imidazole ethylnicotinic acid. The *OsDST* gene, engineered in the rice cultivar MTU1010 via CRISPR-Cas technology, exhibited augmented tolerance to both drought and salt stress, primarily due to increased leaf retention under drought conditions [269]. Furthermore, genetic engineering of genes such as *OsBADH2*, *OsMPK2*, *SAPK2*, and *OsPDS* has demonstrated potential for enhancing abiotic stress tolerance in rice [270]. Gene editing of *OsHAK1*, *OsNramp5*, and *OsARM1* in rice has resulted in a reduced accumulation of arsenic, cadmium, and calcium, respectively [271]. Furthermore, [272] showed that CRISPR-mediated editing of the *OsRAV2* gene in rice improved salt stress tolerance. Some other research has utilized CRISPR-Cas9 to develop rice mutants resistant to salt stress by targeting *OsDST* [273], *OsNAC45* [114], *AGO2* [274], and *OsBBS1* [275]. Table 3 highlights innovative strategies for rice microbiome engineering aimed at enhancing rice productivity, stress tolerance, and climate resilience. These strategies include advanced tools such as gene editing, microbial inoculation, traditional rice plant breeding techniques, and high-throughput metagenomic sequencing. By integrating these approaches, the potential for sustainable improvements in rice cultivation to address global agricultural challenges is significantly amplified.

## 12. Microbiome-Shaping (M) Genes: Unlocking New Avenues for Stress-Resilient Traits

The rhizosphere (interface between roots and soil), phyllosphere (surface of leaves), and endosphere (internal tissues) are the three main compartments into which the plant microbiota can be divided [278]. The phyllosphere, or biological niche, which is made up of a variety of microorganisms, is the greatest biological habitat found in plant leaves. Controlling pathogenic bacteria and preserving the host’s overall health depends on the existence of beneficial microbial members in the phyllosphere [194,279]. These beneficial microbes support innate immunity by serving as an additional line of defence [280]. Increasing microbiome homeostasis and manipulating the microbiota specifically have emerged as viable strategies for the long-term preventative suppression of plant diseases [33,281].

Several lines of evidence suggest that rice plant has the ability to modify its microbiome through the expression of particular genes [282,283,284]. According to reports, these genes that shape the microbiome help plants perform better and become more resilient to a variety of stresses. These stresses include defence against soilborne pathogens [285], coping with nitrogen deficiency, enhancing nitrogen acquisition [286], and striking a balance between growth and defence [287]. These genes all have the trait of being changed by the bacteria found in the rhizosphere or roots.

Su et al. [51] observed distinct phyllosphere microbial communities between indica and japonica rice varieties at various taxonomic levels, suggesting a potential influence of host genetics. They found that four bacterial orders (Pseudomonadales, Burkholderiales, Xanthomonadales, and Enterobacterales) were particularly responsive to specific host genetic backgrounds, notably those linked to the phenylpropanoid biosynthesis pathway. Further investigation revealed haplotype differences in the *OsPAL02* gene between indica and japonica, leading to differential 4-hydroxycinnamic acid (4-HCA) production, a key precursor for lignin biosynthesis. Plant-emitted VOCs can shape the phyllosphere microbial community by acting as antimicrobial agents or carbon sources, while the shaped microbial community can in turn influence plant physiology [207]. A recent study has revealed that rice employs a secondary metabolite biosynthetic gene to regulate the balance of its leaf microbiome. This finding suggests a promising avenue for developing disease-resistant crop varieties by manipulating microbiome-influencing genes [288]. This suggests that Ms genes, which are involved in the synthesis or release of secondary metabolites, may be particularly sensitive to environmental changes. This sensitivity could allow plants to modulate their interactions with beneficial microbes, thereby enhancing their ability to adapt to changing environmental conditions [242].

All of these data point to the likelihood that the genes that shape the microbiome are spread and preserved in plant hosts, which were defined as “M genes” in [146]. In general, the use of M genes in crop cultivars that are naturally enriched in useful microbial taxa may provide new opportunities for the molecular breeding of desirable traits [289]. The M genes allow for direct disease-suppressive effects as well as the enrichment of disease-suppressive microbiota in the context of disease-resistant breeding. Rapidly evolving diseases are posing an increasing challenge to current breeding tactics, which rely on the accumulation of resistance genes [289]. However, a different kind of breeding relies on modifying the host’s susceptibility genes. On the other hand, deploying M genes offer a strategic approach to directly influence and manipulate the host-microbiota relationship. This strategy might make it easier to produce crop varieties with longer-lasting disease resistance and a wider range of applications. Understanding how the M genes work in connection with the host plant’s innate immunity to maintain microbiome homeostasis is still developing, even though the notion of the M gene has filled a knowledge gap regarding the interactions between pathogens, host plants, and the resident microbiota [146]. Using gene editing tools (such CRISPR-Cas9) to alter or incorporation of the M genes to the crop genome could be a rapid strategy to develop disease resistant cultivars. The discovery of M genes in various plants, including rice genotypes, presents an intriguing resource for engineering microbiomes through molecular breeding. This approach aims to maximize the benefits derived from the associated microorganisms. Further research is essential to elucidate the role of M genes in rice genotypes.

## 13. Overcoming Challenges and Exploring Future Prospects in Rice Microbiome Engineering

Microbial communities associated with rice are dynamic and influenced by various environmental factors such as soil type, climate, and plant genotype [290,291]. Notably, the rice microbiome is primarily shaped by the microhabitat rather than geographic location, emphasizing the need for targeted interventions that consider the specific growing environments of rice [146]. Core endophytes, including *Pantoea* and *Xanthomonas*, play vital roles in nutrient uptake and stress tolerance; however, their capacity to antagonize pathogens is generally limited [146]. Moreover, different rice genotypes display distinct root microbiome architectures, which significantly influence plant growth and stress responses [292]. The rhizosphere of rice supports a complex ecosystem of microorganisms with intricate interactions, making understanding and manipulating these relationships a prominent challenge in microbiome engineering [114,293].

One of the key challenges is predicting and managing the dynamics of these microbial communities. While major rhizobacterial phyla can be cultivated for genome sequencing and phenotypic characterization, it remains difficult to assign specific functional roles to individual microbes or groups due to within-species variability. The study of plant–microbial interactions is further complicated by the predominance of unculturable microorganisms, which may harbour untapped potential for beneficial applications in sustainable agriculture, highlighting a relevant area for continued exploration.

Numerous methodologies have been developed to study soil microorganisms, including the traditional approach of cultivating microbes on solid or liquid media. This method allows for the isolation of specific microbial strains for biochemical and physiological evaluation. Recent advancements in culture-based techniques, such as dilution-to-extinction culturing, have contributed to a growing inventory of known soil microbial isolates, facilitating the identification of hundreds of heterotrophic bacterial taxa. Additionally, cutting-edge high-throughput cultivation methods, termed ‘culturomics’, harness machine learning and robotics to accelerate the isolation of microorganisms from microtiter plates [294].

Metagenomic techniques hold significant promise but also face limitations in accurately characterizing microbial communities within specific soil types and in detecting ecologically important, low-abundance species [295]. While DNA sequencing, such as 16S rRNA amplicon sequencing for bacteria and archaea, and 18S rRNA for eukaryotes, provides a comprehensive view of the soil microbiome—capturing both active and dormant cells—this method tends to include dead cells and primarily identifies taxonomic signatures without detailing their functional potential [296]. Although high-throughput sequencing technologies ameliorate some depth-of-coverage challenges, they also impose substantial computational requirements for managing extensive datasets. Quantitative analyses become problematic without absolute abundance data, as many sequencing studies predominantly rely on relative abundance, making it difficult to discern true abundance shifts within communities. Additionally, the instability of RNA, combined with the complexities of extracting it from soil, presents further obstacles, as does the pre-eminence of ribosomal RNA that can obscure the sequencing of messenger RNA transcripts, which are critical for understanding functional dynamics. Inefficient protein extraction hinders the recovery and subsequent analysis of proteins expressed by soil microbial communities, thus limiting the depth of metaproteomic studies. However, emerging bioinformatics tools—such as DRAM, MEMPIS, and XCMS—provide promising avenues for the analysis and integration of omics datasets, enhancing our ability to identify specific species, genes, enzymes, and functions within the soil microbiome that could be leveraged for engineering applications.

Our understanding of the complex interactions between host plants, pathogens, and their associated microbiota is still evolving, particularly concerning the role of M genes and their interplay with the host plant’s innate immunity. These genes not only regulate metabolite levels but also influence overall microbiome composition, with the potential to enhance crop yield and disease resistance. However, future climate change, especially intensified drought conditions, poses significant challenges to plant–soil microbial communities and their intricate interactions. Drought disrupts the activity of beneficial microorganisms, reducing their ability to sustain ecosystems and support plant growth, while simultaneously increasing the activity and infectious potential of plant pathogenic microbes [297]. Sustainable agronomy under such conditions will rely on adaptive strategies and ecological balance within soil–plant communities. Furthermore, a recent comprehensive review by Loiko and Islam [298] highlights that climate change (e.g., drought) may accelerate horizontal gene transfer (HGT) among soil–plant microbes, enhancing the spread of pathogenic traits with critical implications for plant health and ecosystem resilience. Despite these challenges, targeted microbiome engineering offers promising solutions to foster sustainable agricultural practices by strengthening beneficial plant–microbe relationships. Investigations into the rice microbiome, for example, have the potential to reveal novel biofertilization strategies and improve stress resilience. Continued research should prioritize understanding microbial interactions, identifying new beneficial strains, and integrating microbiome engineering with traditional breeding techniques to enhance stress resilience and nutritional efficiency in crop varieties.

## 14. Conclusions

The multifaceted relationships between rice plants and their associated microbial communities serve as a cornerstone for developing sustainable rice production practices capable of addressing the pressing challenges of food security and climate resilience. This comprehensive review highlights the significant roles of diverse microbial populations including plant growth-promoting probiotic bacteria, fungi, and endophytes in enhancing rice health, yield, and resilience against both biotic and abiotic stresses. Through the manipulation and engineering of the rice microbiome, it is possible to optimize plant growth and improve nutrient uptake mechanisms while reducing reliance on harmful chemical inputs, thereby fostering healthier agroecosystems. Recent advancements in metagenomics and other omics technologies have opened new avenues for understanding the complexities and dynamics of rice-associated microbial communities. These technologies afford unprecedented insights into the gene–microbe interactions that underpin plant health and productivity. By identifying and leveraging beneficial microbial taxa along with their microbiome-shaping (M) genes, researchers can develop innovative biotechnological strategies aimed at reinforcing host plant resilience, enhancing nutrient acquisition and use efficiency, and improving overall crop performance in a sustainable manner. While significant progress has been made, ongoing challenges remain in accurately predicting and managing the dynamics of these microbial communities, considering factors such as environmental variability and plant genotype influences. Future studies should prioritize characterizing the functional capabilities of these microbiomes, exploring novel microbial associations, and integrating microbiome engineering with traditional breeding efforts to boost rice productivity. As we move towards an era of precision agriculture, harnessing the potential of rice-associated microbes will be crucial in developing resilient rice varieties capable of thriving in an unpredictable climate, ultimately contributing to sustainable agricultural practices that ensure food security for future generations. The path forward necessitates a collaborative approach that combines the expertise of plant biologists, microbiologists, geneticists, computational biologists, and agronomists to refine our understanding of the complex interactions that define plant–microbe ecosystems. Emphasizing practical applications and real-world scalability will be key in translating laboratory findings into effective strategies that can be implemented in diverse agricultural contexts. Fostering synergistic relationships between rice plants and beneficial microbes can pave the way for sustainable agriculture that aligns with ecological principles and addresses the challenges of climate change. Microbiome engineering has the potential to enhance plant–microbe interactions, improving resilience to biotic and abiotic stresses while boosting crop yields. Future efforts should focus on understanding microbial dynamics and integrating these innovations with breeding to ensure long-term agricultural sustainability.

## Figures and Tables

**Figure 1 microorganisms-13-00233-f001:**
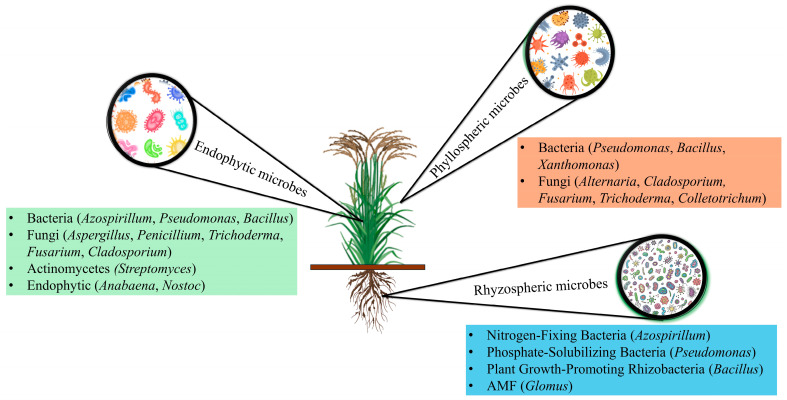
Major types of rice plant microbiomes. The diagram highlights endophytic (within tissues), phyllospheric (on leaves), and rhizospheric (at the root–soil interface) microbes, each contributing to plant health and stress resilience.

**Figure 2 microorganisms-13-00233-f002:**
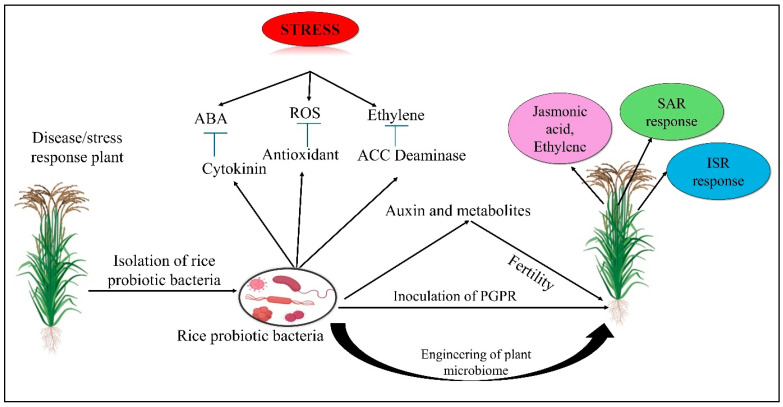
Microbial involvement in abiotic/biotic stress tolerance in rice. Abbreviations: ABA (Abscisic acid), ROS (reactive oxygen species), SAR (systemic acquired resistance), ISR (induced systemic resistance), ACC (1-Aminocyclopropane 1-carboxylic acid).

**Figure 3 microorganisms-13-00233-f003:**
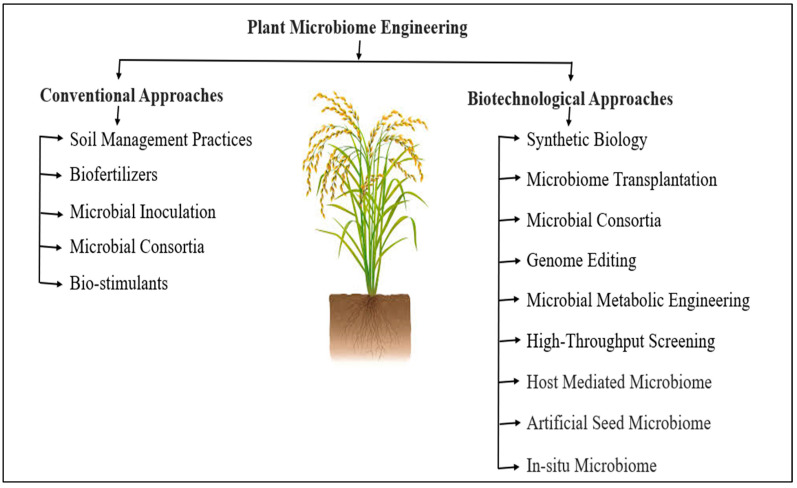
Approaches to microbiome engineering in rice (including conventional and biotechnological methods).

**Table 1 microorganisms-13-00233-t001:** Role of beneficial microbes in controlling rice diseases.

Beneficial Antagonistic Microbes	Phytopathogens	Observed Effects	References
*Bacillus amyloliquefaciens* and *Aspergillus spinulosporus*	*Xanthomonas oryzae* pv. *oryzae*	Increase the expression of defence-related enzymes, and proteins, and elevate levels of total phenols	[167]
	*Curvularia lunata*, *Fusarium semitectum*, and *Helminthosporium oryzae*	Suppress the germ tube elongation and mycelial development in fungal infections	[168]
	*Acidovorax oryzae*	Cell membrane damage results in decreased cell count, biofilm development, and impaired swimming capability	[169]
Consortium of *S. fimicarius*, *S. laurentii*, *P. putida*, and *Metarhizium anisopliae*	*X. oryzae* pv.* oryzae*	Decrease the occurrence of leaf blight	[170]
*B. subtilis*, *B. amyloliquefaciens*, and *B. methyltrophicus*	*X. oryzae* pv. *oryzae*	Activate the defence-related enzymes	[171]
*P. aeruginosa*	*X. oryzae* pv. *oryzae*	Activates the defence enzymes	[172]
*Streptomyces* spp.	*B. glumae*	Inhibits the growth of *B. glumae* and promotes plant growth	[173]
*P. fluorescens*	*M. oryzae*	Reduces the physical damage caused by *M. oryzae*	[174]
*Glomus intraradices*	*M. oryzae*	Enhances the expression of defence-related genes (*OsNPR1*, *OsAP2*, *OsEREBP*, and *OsJAmyb*)	[175]
*Talaromyces* spp.	*R. solani*	Increases the expression of defence-related genes and defence enzymes synthesis	[176]
*Serratia marcescens*	*R. solani*	Decreases the occurrence of sheath blight	[177]
*P. fluorescens*	*P. oryzae*	Induces integrated stress response (ISR) in rice against *P. oryzae*	[178]
*Bacillus subtilis*	*R. solani*	Reduces the frequency of sheath blight	[179]
	*R. solani*	Decreases the occurrence of sheath blight and enhanced plant growth	[180,181]
*Bacillus subtilis*	*M. oryzae*	Reduces (by over 50%) blast disease, enhances systemic resistance, improves plant resilience	[88,143,145,182,183]
*B. oryzicola*	*Gibberella fujikuroi*	Decrease Bakanae disease severity by 46–78%.	[184]
	*B. glumae*	Stimulates the resistance and enhancement of plant development	[185]
*Streptomyces* spp.	*M. oryzae*	Enhances the defensive enzyme activity	[186]
*Cladosporium cladosporioides*	*M. oryzae*	Increase the enzyme activity and expression of defence-related genes like *JIOsPR10*, *LOX-RLL*, and *PR1b*	[187,188]
*B. subtilis*, *B. amyloliquefaciens*, and *B. methyltrophicus*	*X. oryzae* pv. *oryzae*	Activate ISR leads to increase the activity of defence-related enzymes	[172]
*P. aeruginosa*	*X. oryzae* pv. *oryzae*	Increases the functions of defence-associated enzymes	[173]
*Streptomyces* spp.	*B. glumae*	Suppresses *Burkholderia glumae* development	[174]
*Consortium of S. fimicarius*, *S. laurentii*, *P. putida*, and *Metarhizium anisopliae*	*X. oryzae* pv. *oryzae*	Decrease the occurrence of leaf blight	[171]
*Bacillus thuringiensis*	*Scirpophaga incertulas*	Reduces pest damage, enhances plant growth and yield	[189]
*Pseudomonas fluorescens*	*X. oryzae* pv. *oryzae*	Reduces bacterial blight, promotes plant growth, increases disease resistance	[153,154]
*Azospirillum brasilense*	*M. oryzae* *X. oryzae*	Enhances growth, induces resistance against multiple pathogens, improves disease tolerance	[165]
*Bacillus velezensis*	*P. oryzae* *Bipolaris oryzae*	Reduces fungal disease incidence, enhances plant growth, improves grain yield	[166]
*B. megaterium*	*R. solani*	Suppresses sheath blight, promotes plant growth, increases disease resistance	[166]
*B. toyonensis*	*B. oryzae*	Suppresses sheath blight, promotes plant growth, increases disease resistance	[166]
*Bradyrhizobium japonicum*	*Ralstonia solanacearum*	Controls bacterial wilt, enhances plant health and disease resistance	[166]
*Trichoderma* spp.	*R. solani* *P. oryzae* *Fusarium* spp. *B. oryzae*	Suppresses fungal pathogens, promotes growth, reduces disease incidence	[155,156,157]
*T. asperellum*	*R. solani*	Suppresses fungal growth, reduces disease severity	[163]
*Lactobacillus* spp. and *Aspergillus* spp.	*Ustilaginoidea virens*	Reduces pathogen infection and disease severity in rice panicle	[190]
*Azospirillum* spp.	Various soilborne pathogens	Increases rice growth and yield, enhanced stress resistance	[191,192]
*Saccharothrix* spp.	*P. oryzae*	Produces host plant beneficial bioactive compounds	[193]

**Table 2 microorganisms-13-00233-t002:** Microbes contributing to the improved tolerance to abiotic stresses in rice.

Beneficial Microbes	Mechanism	References
*Trichoderma harzianum*	Improves root development in water scarcity like salinity stress	[200]
	Increases the expression of aquaporin, dehydrin, and malondialdehyde genes, as well as other physiological factors	[201]
	Improves seed germination and seedling growth at different stress conditions decreasing oxidative damage and lipid peroxidation	[202]
	Enhances the phenol levels, peroxidase activity, lignin content, and cell membrane integrity	[203]
	Increases the levels of antioxidant enzymes and secondary metabolites in plants	[204]
	Improves gene expression associated with stress response	[199]
	Enhances the efficiency of photosynthetic, antioxidant enzymes, and physiological adaptation in saline environments	[205]
*Pseudomonas pseudoalcaligenes* and *Bacillus pumilus*	Decrease the toxicity of reactive oxygen species (ROS)	[206]
	Increase the amount of osmoprotectants in rice, like glycine betaine-like quaternary compounds, to help shoots grow more when they are under saline stress	[207]
	Inhibit the absorption of Na^+^ ions Synthesize growth related metabolites and enzymes	[123,208]
	Reduce sodium uptake in roots under saline stress conditions	[209]
	Protect cells from saline stress	[210]
	Enhance the synergistic interaction among several PGPR strains	[210]
	Reduce abiotic stress by increasing plant hormone, osmolytes, antioxidants, and growth-regulated genes	[211]
*B. amyloliquefaciens*	Improves photosynthesis, hormone signalling, stress response, and carbohydrate metabolism	[212]
	Enhances the synthesis of secondary metabolites, hormones, and enzymes	[213,214]
	Increases the production of indole-3-acetic acid, siderophores, and cellulase under stress condition	[214]
	Increases biomass, relative water, and proline content under stress conditions	[212,215]
*Brevibacterium* sp.	Increases tolerance to salinity stress	[216]
	Increases the expression of stress linked genes	[217,218]
	Reduces ethylene release and reactive oxygen species levels in rice	[217]
	Decreases arsenic absorption in rice plants and lower stress-related enzyme activity	[218]
*Bacillus* sp.	Enhances levels of phenylalanine ammonia lyase, peroxidase, and polyphenol oxidase to combat bacterial leaf blight	[218]
	Enhances tolerance to water stress	[217]
	Inhibits sodium ion absorption and enhances antioxidant enzyme	[219]
	Enhances resistance to cold and drought stress	[214]
*Glomus intraradices*	Increases resistance to rice blast disease and improves phosphorus nutrition	[220]
AMF	Reduces Cd uptake in rice and improves micronutrient (Zn and Fe) under flooding conditions	[221]
*Funneliformis mosseae* + *Piriformospora indica*	Increase salinity tolerance	[220]

**Table 3 microorganisms-13-00233-t003:** Engineering rice microbiome using gene editing, inoculation, traditional breeding and metagenomic approaches for crop improvement.

Approach	Method	Outcome	References
CRISPR-Cas9 Gene Editing	Gene editing to enhance disease resistance	CRISPR/Cas9 protocol for genome editing of *Pyricularia oryzae* (rice blast fungus). Enables gene disruption, base editing, and functional genomics	[256]
	Bacterial blight resistance	Developed bacterial blight-resistant rice by silencing *OsSWEET11*, *OsSWEET13*, and *OsSWEET14* genes, which regulate sugar transport in the plant	[276]
	Resistance enhancement via CRISPR	Improved rice resistance to *Magnaporthe oryzae* by editing the *OsHDT701* gene	[260]
	Blast disease resistance	Amino acid substitution in *ALS* gene of basmati rice significantly improves resistance to bacterial blight	[258]
	CRISPR editing for enhanced stress tolerance	Improved drought and salt stress tolerance in rice by editing the *OsDST* gene	[270]
	Salt and abiotic stress tolerance	Improved salt stress tolerance in rice via CRISPR editing of *OsRAV2* and *OsDST* genes	[114,273]
	Reduced heavy metal accumulation	Reduced arsenic, cadmium, and calcium accumulation in rice by editing *OsHAK1*, *OsNramp5*, and *OsARM1* genes	[271]
Microbial Inoculation	Beneficial microbial strains	Introduction of beneficial microbes (e.g., *Pseudomonas*, *Bacillus*) to enhance nitrogen fixation and suppress pathogens in rice	[5]
	Endophytic microbial engineering	Modification of endophytic bacteria to promote plant health and stress tolerance by enhancing the plant microbiome	[255]
	Rhizosphere microbial community manipulation	Modulation of the plant rhizosphere microbiome to improve disease resistance and nutrient uptake in rice	[260]
Synthetic Biology	Engineering synthetic microbial consortia	Design and application of synthetic microbial communities to optimize plant–microbe interactions and improve plant resilience	[256]
	Microbe-engineered growth-promoting substances	Engineering microbes to produce beneficial compounds (e.g., antimicrobial peptides, growth regulators) to enhance plant health	[5]
Traditional Breeding	Selection for microbiome-supportive traits	Traditional breeding to select rice varieties that support beneficial microbial communities through exudate production or root architecture	[268]
	Breeding for enhanced plant–microbe interactions	Breeding rice varieties with traits that favour beneficial plant–microbe interactions, such as improved exudate profiles that attract beneficial microbes	[265]
Metagenomics/Microbiome Profiling	High-throughput sequencing of microbiomes	Profiling rice microbiomes to identify beneficial microbes and determine how plant varieties impact microbial communities.	[268]
	Microbial community analysis and optimization	Metagenomic analysis to identify microbial communities that enhance plant resilience to stresses like drought and disease	[5]
Environmental Modification	Soil amendments to enhance microbial diversity	Use of biochar, organic fertilizers, and other soil amendments to promote beneficial microbiomes in the rhizosphere of rice	[274]
	Fertilizer application to modulate microbiome	Modulation of the plant microbiome through strategic fertilizer application, enhancing nutrient availability and plant health	[277]

## Data Availability

The original contributions presented in the study are included in the article, further inquiries can be directed to the corresponding authors.

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
