# Peer review of "Microbiome Engineering for Sustainable Rice Production: Strategies for Biofertilization, Stress Tolerance, and Climate Resilience"

_microorganisms, 2025, doi:10.3390/microorganisms13020233_

Round 1
Reviewer 1 Report
Comments and Suggestions for Authors
Minor comments:
Line 95-106: Explain, what are the mechanisms by which biopolymers and osmolytes enhance stress tolerance in rice, and how might these be applied more broadly in crop management.
Line 146-163: How effective are microbiome engineering strategies like gene editing and microbial inoculation in developing stress-resistant rice varieties, and what challenges remain in their application? Explain in brief
Line 173-188: To what extent do planting location and soil type versus rice genotype contribute to the composition of rhizosphere microbial communities?
Line 327-343: How do endophytic microorganisms enhance rice stress tolerance and growth without inducing pathogenic symptoms, and what implications does this have for crop breeding programs?
Line 363-382: What roles do microbial signaling molecules such as AHLs and VOCs play in the rhizosphere, and how do they influence rice plant health and nutrient acquisition?
Line 475-486: How do microbes like PGPRs contribute to rice yield improvement through mechanisms other than nitrogen fixation, and what are the potential limitations of these approaches?
Line 575-585: How do microbes mitigate abiotic stresses like salinity and drought in rice, and what advancements are needed to harness these benefits on a larger scale?
Comments on the Quality of English LanguageMinor comments
Author Response
Reviewer 1 (revisions can be seen in the track-change)
Reviewer’s comment: Line 95-106: Explain, what are the mechanisms by which biopolymers and osmolytes enhance stress tolerance in rice, and how might these be applied more broadly in crop management.
Our Response: We have revised the manuscript about the mechanism of biopolymers and osmolytes enhance stress tolerance in rice and how these might be applied more broadly in crop management.
Reviewer’s comment: Line 146-163: How effective are microbiome engineering strategies like gene editing and microbial inoculation in developing stress-resistant rice varieties, and what challenges remain in their application? Explain in brief
Our Response: Microbiome engineering is an emerging topic. Modulation in the microbiome by inoculation of beneficial microorganisms or engineering the genomes of the existing microbiome are considered a new way for adding beneficial effects to cultivated rice. We added an explanation with suitable references.
Reviewer’s comment: Line 173-188: To what extent do planting location and soil type versus rice genotype contribute to the composition of rhizosphere microbial communities?
Our Response: It is an interesting question. However, the extent is not precisely understood. However, both geneotype and planting location have an effect on the microbiome.
Reviewer’s comment: Line 327-343: How do endophytic microorganisms enhance rice stress tolerance and growth without inducing pathogenic symptoms, and what implications does this have for crop breeding programs?
Our Response: Endophytic microorganisms enhance rice stress tolerance in various ways, including production of metabolites, including phytohormones, induction of host gene expression, fixation of atmospheric nitrogen, etc.
Reviewer’s comment: Line 363-382: What roles do microbial signalling molecules such as AHLs and VOCs play in the rhizosphere, and how do they influence rice plant health and nutrient acquisition?
Our Response: This is a valid question. Many VOCs play roles in suppressing pathogens and inducing systematic resistance.
Reviewer’s comment: Line 475-486: How do microbes like PGPRs contribute to rice yield improvement through mechanisms other than nitrogen fixation, and what are the potential limitations of these approaches?
Our Response: The PGPR significantly promotes rice growth by the production of phytohormones, solubilisation of essential nutrients, and regulation of gene expression in the host.
Reviewer’s comment: Line 575-585: How do microbes mitigate abiotic stresses like salinity and drought in rice, and what advancements are needed to harness these benefits on a larger scale?
Our Response: As mentioned earlier, microbes produce diverse metabolites and enzymes and induce gene expression to help host stress tolerance.
Reviewer 2 Report
Comments and Suggestions for Authors
The review manuscript by the authors explained that the plant microbiome, encompassing the rhizosphere, phyllosphere, and endosphere, plays a critical role in nutrient acquisition, stress tolerance, and overall plant health. While extensive research exists on crop microbiomes, our meta-survey identified a significant gap in focused reviews addressing the rice plant microbiome. As the staple food for over half the global population, rice faces mounting challenges from climate change and environmental stresses. This review bridges this knowledge gap by synthesizing advancements in understanding rice-associated microbiomes' diversity and functional roles. It highlights their contributions to plant growth, nutrient uptake, and resilience against biotic and abiotic stresses, emphasizing rhizosphere signaling mechanisms that mediate plant-microbe interactions. Advances in metagenomics have transformed our ability to map rice microbiomes, uncovering their composition and functional potential. The review also explores microbiome engineering strategies to develop stress-resistant rice varieties, including microbial inoculation, gene editing, and microbiome-shaping (M) genes. These tools promise to improve rice productivity, reduce chemical inputs, and enhance climate resilience. However, challenges remain in deciphering microbial interactions, environmental variability, and scalable application of microbiome technologies. This review provides critical insights into leveraging microbiomes for sustainable rice production and outlines future directions for integrating advanced genomic tools and microbiome engineering into global agricultural practices.
The abstract and conclusions should be incorporated with the additional information about how these microbiomes could work in long-term climatic changes.
Another major concern is that the figure legends must be explained in more detail.
Author Response
Reviewer 2 (revisions can be seen in the track-change)
Reviewer’s comment: The abstract and conclusions should be incorporated with the additional information about how these microbiomes could work in long-term climatic changes.
Our Response: Thank you for your valuable suggestion. We have made brief revisions to both the abstract and conclusion. The microbiome can impact the host plant's plasticity, thereby influencing its phenology under changing environmental conditions. Additionally, we have included new information in Section 13 (Overcoming Challenges and Exploring Future Prospects in Rice Microbiome Engineering) based on recent research, which explores the effects of future global climate changes, such as drought, on soil-plant microbial communities and their interactions. Please see 1012-1028.
Reviewer’s comment: Another major concern is that the figure legends must be explained in more detail.
Our Response: We revised the figure legends.
Reviewer 3 Report
Comments and Suggestions for Authors
This review article addresses a significant problem for agricultural praxis. The article is based on more than 200 cited works. It addresses a very broad aspect of the rice microbiome. It was written as a result of a great deal of time and effort. This article should be published in Microorganisms. However, at this stage, this manuscript requires a major revision. Detailed comments are provided in Remarks. The revision should cover the following aspects:
- there are many typographical errors in the manuscript
- in several chapters the text does not correspond to the titles of these chapters
- the authors use different terms, but it is not clear whether they understand their meaning correctly. It would be good if they defined them precisely at the beginning of the manuscript (some are explained only at the end of the manuscript).
- The authors should explain in the introduction what microorganisms they are dealing with in this manuscript and should justify the point of view they have adopted. Many fragments mention bacteria, but a species of fungus appears sporadically (without indicating that it is a fungus). For example, the Introduction lists many genera of bacteria, and only once does a fungus appear (Trichoderma).
- The text should be checked for consistency with the cited literature - I have found fundamental errors
Remarks
Line 41 must be defined in the Introduction microbiomeAbstract
Line 45-46 "our meta-survey identified a significant gap in focused reviews addressing the rice plant microbiome", Introduction Line 151-152 "Extensive studies and reviews have explored the rice microbiome, focusing on its diverse functions and potential applications in sustainable agriculture [46-48]." Question - don't these contents contradict each other ???
Line 70-71 "Plant-associated microbiota includes epiphytes on the surface, endophytes in plant tissues, and the rhizosphere, endosphere, and phyllosphere environments" - this text is incorrect, correction is needed. i) endophytes are of course inside plant tissue but this is not a full definition of this term, ii) microbiota includes environments ???
Line 100-101 Compant et al. [30] identified that Sphingomonas melonis naturally confers resistance to seed-borne pathogens like Burkholderia plantarii in rice. - there is no such data in this publication. In addition, the bibliographic data do not match (see Remarks Line 1044-1045.
Line 109 AM plants – requires explanation (arbuscular mycorrhizae?)
Line 110 sp. - it should not be italic (this applies to the entire manuscript)
Line 131 Rhizobium – it should be italic (this applies to genera and species in the entire manuscript
Line 186 this text requires correction (dot)
Line 190-199 here are 3 publications listed with information on what was dealt with in them. Can such a form help to highlight the problem defined in the title of chapter 2: "Understanding the Diversity and Functional Dynamics.." . I would expect synthetic results of these works.
Line 190 three different root niches – this text is unfinished – what niches are you talking about?
Line 218-219 – in my opinion this text is incorrect. The authors must define the individual terms used in the text, as I have already indicated
Line 202 here the term Chloroflexota is used, while in Line 212 and 224 Chloroflexi.
It should be explained why different names are used
Line 221 – 224 this text is unclear
Line 226-227 The phyllosphere community is mainly composed of Proteobacteria, Bacteroidetes, Firmicutes, and Actinomycetes [73] – this is inconsistent with Figure 1, which also lists fungi
Line 229 Figure 1. Please note that in Figure 1 there is endophytic microbes, while in the explanation it is written endospheric. Are endophytic and endospheric synonyms? Line 229 Figure 1 Phyllospheric microbes – errors should be corrected
Line 230 chapter 3 “The Role of Rhizospheric Microbes..” – in this chapter there is too little about the role of…, while other aspects prevail
Line 231-232 the appropriate publication, the author of this definition should be cited
Line 237 and mycorrhizae – it should be clarified what it is about
Line 261-262 the text requires clarification
Line 281 4. Phyllospheric Microbes and Their Contributions to Rice Growth and Disease Resistance – the text of this chapter is not consistent with the title, there is very little about “Contribution….”
Line 301 Candida koribacter – I know that this is the case in the cited publication, but it should be borne in mind that the genus Candida is a representative of fungi
Line 303 REE – this requires explanation
Line 315 12 genera – specify which microorganisms are meant
Line 325 endophytes, … the endospheric microbiome – as indicated above, it should be defined whether these are synonyms, and the basis for the current treatment of these terms should be provided
Line 336- 337 – this text requires minor corrections
Line 342-343 this text is unclear
Line 358-359 the relevant literature should be cited. This is the essence of the problem indicated in the title of chapter 5
Line 394 “Santosh Kumar et al. [95]” – there is an error here, under number 95 in References there is Doni et al. 2022
Line 466 and induced 83.24% resistance against rice – this text is not clear
Line 467 it should be some
Line 472 though evidence in the phyllosphere is limited - please note that the title of chapter 6 concerns the rhizosphere
Line 475 chapter 7 – the text in this chapter does not present the mechanism, which is indicated in the title
Line 546 “This beneficial fungus” – what is ‘this’ supposed to mean
Line 572 Table 1: “Beneficial microbes (bacteria)….”- here is an error that needs to be corrected. This table presents not only bacteria but also antagonistic fungi
Line 572 This table requires correction Talaromyces spp/ Rhizoctonia solani is given twice
Line 617 text requires correction
Line 619 no dot, it should be Curtobacterium sp.,
Line 783 it should be pv. oryzae,
Line 829 this text requires correction
Line 1044-1045 requires explanation ; „Compant, S.; Cassan, F.; Kostić, T.; Johnson, L.; Brader, G.; Trognitz, F.; Sessitsch, A. Harnessing the Plant Microbiome for Sustainable Crop Production. Nat. Rev. Microbiol. 2024, 1–5” - the available bibliographic data show that this article was published in 2025, namely Nature Reviews Microbiology volume 23, pages 9–23 (2025)
Author Response
Reviewer 3 (revisions can be seen in the track-change)
Reviewer’s comment: - there are many typographical errors in the manuscript
Our Responses: Thank you for pointing out the errors for improvement of our manuscript. We have significantly revised the manuscript to remove the typo errors and overall quality of the manuscript.
Reviewer’s comment: - in several chapters the text does not correspond to the titles of these chapters
Our Responses: Thank you for this critical comment. We have substantially revised the manuscript to address your comment.
Reviewer’s comment: - the authors use different terms, but it is not clear whether they understand their meaning correctly. It would be good if they defined them precisely at the beginning of the manuscript (some are explained only at the end of the manuscript).
Our Responses: Yes, we fully agree with your valuable comment. We have defined the terms as you have suggested for improving the readability.
Reviewer’s comment: The authors should explain in the introduction what microorganisms they are dealing with in this manuscript and should justify the point of view they have adopted. Many fragments mention bacteria, but a species of fungus appears sporadically (without indicating that it is a fungus). For example, the Introduction lists many genera of bacteria, and only once does a fungus appear (Trichoderma).
Our Responses: Yes, we agree with you. We mainly focused rice associated bacteria, now we added rice associated fungi and related references. We mentioned this focused in the introduction. Thank you for this important comment.
Reviewer’s comment: The text should be checked for consistency with the cited literature - I have found fundamental errors
Our Responses: We apologize, but we are unsure about the specific inconsistency you mentioned. However, we have thoroughly reviewed the document for any potential issues, such as errors in numbering or formatting or style of writing and have made the necessary corrections. The revised version should be free from the fundamental errors.
Reviewer’s comment: Line 41 must be defined in the Introduction microbiome Abstract
Our Response: Thank you. We have made the revision as per your suggestion.
Reviewer’s comment: Line 45-46 "our meta-survey identified a significant gap in focused reviews addressing the rice plant microbiome", Introduction Line 151-152 "Extensive studies and reviews have explored the rice microbiome, focusing on its diverse functions and potential applications in sustainable agriculture [46-48]." Question - don't these contents contradict each other ???
Our Responses: Thank you for the query. Contradicts in Lines 151-152 are changed to “Several studies and reviews have examined the rice microbiome [51-53]; however, a few have delved into its diverse functions and potential applications in sustainable agriculture under real-world conditions.
Reviewer’s comment: Line 70-71 "Plant-associated microbiota includes epiphytes on the surface, endophytes in plant tissues, and the rhizosphere, endosphere, and phyllosphere environments" - this text is incorrect, correction is needed. i) endophytes are of course inside plant tissue but this is not a full definition of this term, ii) microbiota includes environments ???
Our Responses: Thank you. We have made correction as per your suggestion.
Reviewer’s comment: Line 100-101 Compant et al. [30] identified that Sphingomonas melonis naturally confers resistance to seed-borne pathogens like Burkholderia plantarii in rice. - there is no such data in this publication. In addition, the bibliographic data do not match (see Remarks Line 1044-1045.
Our Responses: Thank you. The reference was mistakenly put. The original reference is Matsumoto et al. [33]. We have corrected it.
Please see line number 145.
Reviewer’s comment: Line 109 AM plants – requires explanation (arbuscular mycorrhizae?)
Our Responses: Thank you. We have explained the mentioned line during revision.
Reviewer’s comment: Line 110 sp. - it should not be italic (this applies to the entire manuscript)
Our Responses: Okay, thank you. We have corrected.
Reviewer’s comment: Line 131 Rhizobium – it should be italic (this applies to genera and species in the entire manuscript
Our Responses: Thank you for mentioning the error in style. We corrected throughout the manuscript where the relevant changes are needed according to the role.
Reviewer’s comment: Line 186 this text requires correction (dot)
Our Responses: We apologise for our mistake. We have corrected the error.
Reviewer’s comment: Line 190-199 here are 3 publications listed with information on what was dealt with in them. Can such a form help to highlight the problem defined in the title of chapter 2: "Understanding the Diversity and Functional Dynamics.." . I would expect synthetic results of these works.
Our Responses: We agree with the reviewer’s comment. We corrected the above-mentioned area by replacing the text. Correction can be seen in the track change.
Reviewer’s comment: Line 190 three different root niches – this text is unfinished – what niches are you talking about?
Our Responses: Thank you. For more clarity we explained the above-mentioned query. Correction can be seen in the track change.
Reviewer’s comment: Line 218-219 – in my opinion this text is incorrect. The authors must define the individual terms used in the text, as I have already indicated
Our Responses: Thank you. Okay, we have corrected the text.
Reviewer’s comment: Line 202 here the term Chloroflexota is used, while in Line 212 and 224 Chloroflexi. It should be explained why different names are used
Our Responses: Thank you. We have modified texts with most recent taxonomic revisions – Chloroflexota. Correction can be seen in the track change
Reviewer’s comment: Line 221 – 224 this text is unclear
Our Responses: Thank you. We cleared text in more details. Correction can be seen in the track change.
Reviewer’s comment: Line 226-227 The phyllosphere community is mainly composed of Proteobacteria, Bacteroidetes, Firmicutes, and Actinomycetes [73] – this is inconsistent with Figure 1, which also lists fungi
Our Responses: We agree with the reviewer’s observation. We have corrected it, correction can be seen in the track change.
Reviewer’s comment: Line 229 Figure 1. Please note that in Figure 1 there is endophytic microbes, while in the explanation it is written endospheric. Are endophytic and endospheric synonyms? Line 229 Figure 1 Phyllospheric microbes – errors should be corrected
Our Responses: We appreciate reviewer for pointing out the mistake. We have corrected the figure 1. Correction can be seen in track change.
Reviewer’s comment: Line 230 chapter 3 “The Role of Rhizospheric Microbes..” – in this chapter there is too little about the role of…, while other aspects prevail
Our Responses: Thank you. We removed this reference.
Reviewer’s comment: Line 231-232 the appropriate publication, the author of this definition should be cited
Our Responses: Appropriate citation added in the revised manuscript. correction can be seen in the track change. Thank you.
Reviewer’s comment: Line 237 and mycorrhizae – it should be clarified what it is about
Our Responses: Thank you for this suggestion. Correction was done in the revised manuscript.
Reviewer’s comment: Line 261-262 the text requires clarification
Our Response: A clarification was done in the revised manuscript. Thank you.
Reviewer’s comment: Line 281 4. Phyllospheric Microbes and Their Contributions to Rice Growth and Disease Resistance – the text of this chapter is not consistent with the title, there is very little about “Contribution….”
Our Response: The microbiome consists of phyllosphere, rhizosphere and endosphere. Although rhizosphere microbiome and endophytes have been studied considerably higher compared to the phyllosphere microbiome of rice. However, recent literature indicates that they also play in protection and fitness of the host plant. You may have a look at this recent article: Sahu, K.P., Kumar, A., Sakthivel, K. et al. Deciphering core phyllomicrobiome assemblage on rice genotypes grown in contrasting agroclimatic zones: implications for phyllomicrobiome engineering against blast disease. Environmental Microbiome 17, 28 (2022). https://doi.org/10.1186/s40793-022-00421-5
Reviewer’s comment: Line 301 Candida koribacter – I know that this is the case in the cited publication, but it should be borne in mind that the genus Candida is a representative of fungi
Our Response: Yes, Candida is a fungal genus. However, Candidatus Koribacter is bacteria under the phylum of Acidobacteria.
Reviewer’s comment: Line 303 REE – this requires explanation
Our Response: Thank you for this suggestion. We briefly explained it.
Reviewer’s comment: Line 315 12 genera – specify which microorganisms are meant
Our Response: Thank you. We have clarified it.
Reviewer’s comment: Line 325 endophytes, … the endospheric microbiome – as indicated above, it should be defined whether these are synonyms, and the basis for the current treatment of these terms should be provided
Our Response: Thank you. We have defined as per your suggestion.
Reviewer’s comment: Line 336- 337 – this text requires minor corrections
Our Response: Thank you. We have made the correction. Please see line number
Reviewer’s comment: Line 342-343 this text is unclear
Our Response: We have tried to clarify the test of these lines.
Reviewer’s comment: Line 358-359 the relevant literature should be cited. This is the essence of the problem indicated in the title of chapter 5
Our Response: Thank you. We have corrected it.
Reviewer’s comment: Line 394 “Santosh Kumar et al. [95]” – there is an error here, under number 95 in References there is Doni et al. 2022
Our Response: The reference number was mistakenly put. It was corrected in corrected in revised manuscript.
Reviewer’s comment: Line 466 and induced 83.24% resistance against rice – this text is not clear
Our Response: More clarification was done in revised manuscript.
Reviewer’s comment: Line 467 it should be some
Our Response: Thank you for this suggestion. More clarification was done in revised manuscript.
Reviewer’s comment: Line 472 though evidence in the phyllosphere is limited - please note that the title of chapter 6 concerns the rhizosphere
Our Response: Thank you. As discussed earlier, we revised this line.
Reviewer’s comment: Line 475 chapter 7 – the text in this chapter does not present the mechanism, which is indicated in the title
Our Response: Thank you for this critical question. Although mechanisms are not fully understood, we added current knowledge on this aspect.
Reviewer’s comment: Line 546 “This beneficial fungus” – what is ‘this’ supposed to mean
Our Response: Thank you. The fungus which exerts beneficial effect on host plant, is known as beneficial. We have revised.
Reviewer’s comment: Line 572 Table 1: “Beneficial microbes (bacteria)….”- here is an error that needs to be corrected. This table presents not only bacteria but also antagonistic fungi
Our Response: Yes, the titled is changed in revised manuscript. Please see line number 763
Reviewer’s comment: Line 572 This table requires correction Talaromyces spp/ Rhizoctonia solani is given twice
Our Response: Thank you for pointing out this error. We have corrected the table.
Reviewer’s comment: Line 617 text requires correction
Our Response: We have corrected it.
Reviewer’s comment: Line 619 no dot, it should be Curtobacterium sp.,
Our Response: Yes, thank you. We have corrected it.
Reviewer’s comment: Line 783 it should be pv. oryzae,
Our Response: Thank you. We have corrected this error.
Reviewer’s comment: Line 829 this text requires correction
Our Response: We have corrected it in the revised manuscript.
Reviewer’s comment: Line 1044-1045 requires explanation ; „Compant, S.; Cassan, F.; Kostić, T.; Johnson, L.; Brader, G.; Trognitz, F.; Sessitsch, A. Harnessing the Plant Microbiome for Sustainable Crop Production. Nat. Rev. Microbiol. 2024, 1–5” - the available bibliographic data show that this article was published in 2025, namely Nature Reviews Microbiology volume 23, pages 9–23 (2025)
Our Response: Thank you so much for noticing this error. We have corrected this citation.
Round 2
Reviewer 3 Report
Comments and Suggestions for Authors
This review article addresses an very important issue in agricultural practice. It addresses a very broad aspect of the rice microbiome. The article is based on more than 300 cited works. It was written after a great investment of time and effort. The first version of this manuscript required extensive revision. The authors took into account all comments presented in my review. This article should now be published in the journal Microorganisms. I only found a few typographical errors (Remarks).
Remarks
Line 77 microbes’ – rather microbes
Table 1 why do you use here the name BradyRhizobium instaed of Bradyrhizobium. The same applies to publication No. 166 in line 3135. In the original title of this publication is Bradyrhizobium.
Line 1800 oryzae - it should be italic
Line 1903 Enterobacterales) – bracket not italic
Author Response
Reviewer’s comment: - Line 77 microbes’ – rather microbes
Our Responses
Corrected and changed can be seen in track change Reviewer’s comment: Table 1: why do you use here the name BradyRhizobium instead of Bradyrhizobium. The same applies to publication No. 166 in line 3135. The original title of this publication is Bradyrhizobium.
Our Responses
Thank you for this critical comment. We have substantially revised the table 1 and whole manuscript to address your comment. Change can be seen in track change.
Reviewer’s comment: - Line 1800 oryzae - it should be italic
Our Responses Corrected and change can be seen in track change Reviewer’s comment: - Line 1903 Enterobacterales) – bracket not italic
Our Responses Corrected and change can be seen in track change